ns **nature communications**

# Pixel-wise programmability enables dynamic high-SNR cameras for high-speed microscopy

Jie Zhang[1,2] ✉, Jonathan Newman[1,2], Zeguan Wang[2,3], Yong Qian[2,3], Pedro Feliciano-Ramos[1,2], Wei Guo[1,2], Takato Honda [1,2], Zhe Sage Chen [4], Changyang Linghu [5], Ralph Etienne-Cummings[6], Eric Fossum[7], Edward Boyden [2,3] & Matthew Wilson [1,2]

High-speed wide-field fluorescence microscopy has the potential to capture biological processes with exceptional spatiotemporal resolution. However, conventional cameras suffer from low signal-to-noise ratio at high frame rates, limiting their ability to detect faint fluorescent events. Here, we introduce an image sensor where each pixel has individually programmable sampling speed and phase, so that pixels can be arranged to simultaneously sample at high speed with a high signal-to-noise ratio. In high-speed voltage imaging experiments, our image sensor significantly increases the output signal-to-noise ratio compared to a low-noise scientific CMOS camera (~2–3 folds). This signal-to-noise ratio gain enables the detection of weak neuronal action potentials and subthreshold activities missed by the standard scientific CMOS cameras. Our camera with flexible pixel exposure configurations offers versatile sampling strategies to improve signal quality in various experimental conditions.

Wide-field fluorescence microscopy enables direct observation of physiological processes, such as cell signaling and local chemical concentrations. In neuroscience, this method has revolutionized the study of the neural basis of behavior by capturing the dynamics of thousands of cells in behaving animals[1]. However, challenges still remain in using functional fluorescence microscopy to sample fast neural dynamics[2–6], such as the membrane potential that reflects neural activity occurring at millisecond timescales. High sampling speeds (1 kHz) of fluorescence indicators increase pixel readout noise and limit the number of flourescence photons integrated during each sampling period. These factors lead to a low signal-to-noise ratio (SNR) for detecting fluorescence activities associated with spiking events and subthreshold activity. Imaging setups can use high excitation power to maintain the SNR, but suffer from many caveats, such as photobleaching, heat, and cytotoxicity.

Currently, experiments at kHz camera speed were conducted only in head-fixed animals using high-performance but bulky microscopes and were limited to a few minutes in duration[5,6], which was insufficient to observe many behavioral states and their associated neural mechanisms.

While many efforts have focused on improving the imaging systems' optics[7,8] and finding denoising algorithmic solutions[9,10], the key trade-offs of speed and SNR are also fundamentally linked to the image sensor. In this work, we examine and address the limitations of SNR at high speed by introducing a pixel design and sampling method. Advancements in sensor technology will complement ongoing innovations in optics and denoising algorithms. Together, they will enhance our fluorescence imaging technology to enable the tracking of neural activity at millisecond resolution across a large number of neurons, over long experimental durations.

[1]Picower Institute for Learning and Memory, MIT, Cambridge, MA, USA. [2]Department of Brain and Cognitive Sciences, MIT, Cambridge, MA, USA. [3]McGovern Institute for Brain Research, MIT, Cambridge, MA, USA. [4]Department of Psychiatry, NYU Grossman School of Medicine, New York, NY, USA. [5]Department of Cell and Developmental Biology, University of Michigan, Ann Arbor, MI, USA. [6]Department of Electrical and Computer Engineering, Johns Hopkins University, Baltimore, MD, USA. [7]Thayer School of Engineering, Dartmouth College, Hanover, NH, USA. ✉e-mail: jzhang41@mit.edu

## Results

A fundamental trade-off exists between a pixel's sampling speed and SNR. Fast sampling speeds lead to high readout noise, shortened exposure time, and fewer collected photons, inevitably lowering the SNR. The CMOS image sensor, which uniformly exposes and samples an array of pixels, is subject to the same SNR and speed limitation. High frame rates result in low SNR, and low frame rates with long pixel exposure lead to signal aliasing (Fig. 1a, b).

### Programmable pixel-wise exposure CMOS image sensor

Despite this trade-off, physiological signals, such as the membrane voltage of a cell soma, are often spatially correlated, and the same signal can be redundantly captured by multiple pixels within a region of interest (ROI). We demonstrate a CMOS image sensor with pixel-wise programmable exposures (PE-CMOS) to take advantage of the highly correlated nature of microscopy scenes. The PE-CMOS permits flexible exposure at each pixel. This feature allows versatile pixel configurations to increase temporal resolution at sampling physiological signals without sacrificing SNR. In one configuration, the PE-CMOS staggered pixels' exposure in time to acquire fast-spiking events at multiple phases (Fig. 1c), resolving temporal details finer than the sampling period and exposure time. Importantly, this increase in temporal resolution is achieved without raising the pixel sampling rate or reducing exposure time, therefore avoiding sacrificing the SNR. In another configuration, the PE-CMOS samples the ROI with pixels at different speeds, capturing high-frequency events (spiking activity) and weak signals (subthreshold potentials) that are difficult to acquire simultaneously at a fixed frame rate (Fig. 1c). The flexible pixel-wise exposure configuration is not achievable in conventional CMOS architecture, as all the pixels must have the same exposure, limited by the frame rate, and are sampled concurrently (global shutter) or sequentially in lines (rolling shutter).

### PE-CMOS circuit architecture

The PE-CMOS sensor design enables pixel-wise programmability without compromising pixel sensitivity, which is determined by the percentage of effective photodiode area within each pixel (fill factor). Prior approaches to achieve pixel configurability required a high number of pixel-level circuits[11–13] (~25 transistors), which occupied valuable pixel area and reduced the photodiode fill factor (38% and 45% for pixel pitches of 12.6 and 11.2 μm, respectively[12,13]). In contrast, the PE-CMOS achieves pixel-wise programmability using only six transistors per pixel, enabling high sensitivity (75% and 52% fill factor for smaller pixel pitch of 10 μm and 6.5 μm), comparable to high-performance CMOS sensors (Fig. 1d, e). For the PE-CMOS with a 10 μm pitch, the measured conversion gain is 110 μV/e-, with a read noise of 2.67 e-, measured at room temperature without active cooling. The quantum efficiency (QE) of the pixel is 68% without on-chip microlens array.

The PE-CMOS chip was fabricated using the 180 nm CMOS image sensor process from XFAB, a commercial foundry service. Two prototype designs were produced with pixel arrays containing a 6.5 μm (Chip A) and 10 μm (Chip B) pixel pitch (Supplementary Fig. 1a and Methods). The pixel array is arranged in rows and columns, with readout circuitry at each column (Supplementary Fig. 1b and Methods). Each column readout circuitry comprises a programmable gain amplifier (PGA), correlated-double sampling (CDS) circuits, and an analog-to-digital converter (ADC). The PGA offers a variable gain of 8 – 64 to the pixel output. The amplified signal is then processed by the CDS circuits to reduce flicker (1/f) noise before being sampled by the 10-bit ADC. The row-wise signals (RST, TX, and SEL) multiplex pixels to the readout circuits at each column. These signals also control pixel reset (RST), charge transfer (TX), and row-wise multiplex (SEL) operations. In a single readout period (0.9 ms in our implementation), each column readout circuit selects pixels of the column that end the

exposure and converts their outputs to digital bits that are then sent off-chip.

The PE-CMOS pixel design (Fig. 1d) comprises 6 transistors (T1–T6) and one pinned photodiode (PD). T1 to T4 form the standard rolling shutter 4T-pixel design[14]. During pixel exposure, the PD converts incoming photons into electrons. When the PD is ready to be sampled, T3, controlled by a row reset (RST) signal, first resets the voltage on the floating diffusion (FD) node. T4 (controlled by TX) then moves the electrons from the PD to FD, producing a change in voltage. The voltage signal is buffered by T1 and connected to the readout circuits by T2 (controlled by SEL), where the column readout circuitry samples it.

We incorporated two additional transistors (T5 and T6) into the PE-CMOS pixel design to enable pixel-level exposure programmability. T5 and T6 operate as switches with input driven by column signals, EX. In PE-CMOS, each set of row signals (TX, RST, SEL) selects K rows of pixels as candidates for readout ($K = 8$ in Chip A, $K = 4$ in Chip B). At each column, out of these K pixels, only the pixel with both T5 and T6 activated will end its exposure phase and be sampled by the column circuitry. Meanwhile, the other pixels, with their T5 and T6 remaining off, will continue their exposure. T5 and T6 are controlled by column bus lines EX < 1:K> placed at each column. Pixel-wise exposure control can be achieved by synchronizing the EX signals with row signals (RST, SEL, and TX). An example pixel-wise operation diagram is shown in Supplementary Fig. 1d. For a group of K pixels, the pixel ends its exposure (at the time marked by dotted red line in Supplementary Fig. 1d) whenever its corresponding EX signal is high during the readout operation initiated by the signals: RST < N > , TX < N> and SEL < N>.

### Flexible pixel-wise configuration

We can configure the pixels into custom exposure and sampling patterns by controlling the EX-signals at each pixel column, accomplished through a high-speed I/O interface on the PE-CMOS chip. Supplementary Fig. 2 illustrates various examples of pixel arrangements, with each color representing a unique temporal configuration. In a tiled spatial arrangement, pixels of different configurations are uniformly arranged in $2 \times 2$ windows, enabling uniform sampling of the entire array across four distinct temporal configurations (Supplementary Fig. 2a, b). Alternatively, arranging pixels of different exposures in a random spatial configuration can maximize the incoherence property desirable in a compressed sensing imaging framework[15]. Pixel configurations can be updated instantaneously via on-chip control. This enables the optimization of local spatiotemporal resolution and SNR in real-time. For example, the pixels can be organized in ROI-dependent spatial patterns (Supplementary Fig. 2a). Since many fluorescence proteins are predominantly expressed on the cell membrane, we can encircle the bright outer contour of the cell ROI with fast pixels with short exposure times and high sampling rates while employing slower pixels in the dimmer regions of the ROI to enhance the SNR.

### Sampling voltage signals at high temporal resolution with phase-shifted long exposure pixels

To demonstrate that the PE-CMOS can sample a high-temporal resolution voltage signal with low-speed pixels at different phase-offset, we compared PE-CMOS's performance with a scientific CMOS camera (Hamamatsu Orca Flash 4.0 v3 sCMOS). Both sensors were used to simultaneously image (through a 50/50 beamsplitter) the spontaneous activity of cultured mouse hippocampal neurons expressing the ASAP3 genetically encoded voltage indicator (GEVI)[3] (Fig. 2). We set the sCMOS camera to have a 1.25 ms exposure duration, which also determines its temporal resolution and sampling rate (Fs) of 800 Hz. In the PE-CMOS camera, we used a longer exposure time of 5 ms to integrate over the full spike width (half-width of the action potential generated by ASAP3 spikes is ~ 6 ms). We shift pixel exposures at phase offsets in multiples of 1.25 ms (Fig. 1c). This sets the PE-CMOS temporal

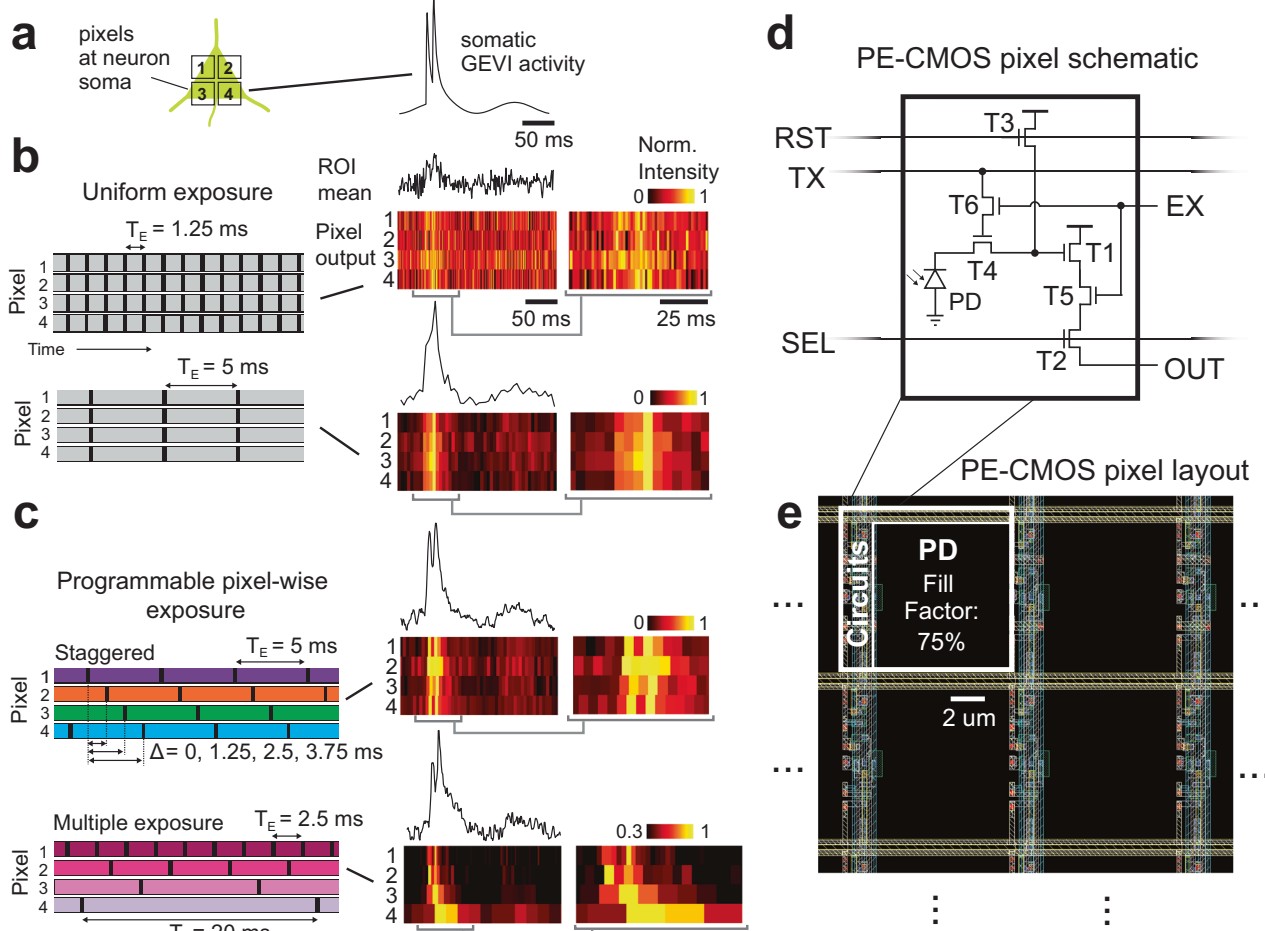

**Fig. 1 | Programmable pixel-wise exposure enables flexible pixel configuration to sample an ROI with high SNR and temporal resolution. a** Pixels within an ROI capture spatiotemporally-correlated physiological activity, such as signals from somatic genetically encoded voltage indicators (GEVI). **b** Simulated CMOS pixel outputs with uniform exposure ($T_E$) face the trade between SNR and temporal resolution. Short $T_E$ (1.25 ms) provides high temporal resolution but low SNR. Long $T_E$ (5 ms) enhances SNR but suffers from aliasing due to low sample rate, causing spikes (10 ms interspike interval) to be indiscernible. Pixel outputs are normalized row-wise. Gray brackets: the zoomed-in view of the pixel outputs. **c** Simulated pixel outputs of the PE-CMOS. Pixel-wise exposure allows pixels to sample at different speeds and phases. Two examples: in the staggered configuration, the pixels

sample the spiking activity with prolonged $T_E$ (5 ms) at multiple phases with offsets of ($\Delta = 0, 1.25, 2.5, 3.75$ ms). This configuration maintains SNR and prevents aliasing, as the interspike interval exceeding the temporal resolution of a single phase is captured by phase-shifted pixels. In the multiple exposure configuration, the ROI is sampled with pixels at different speeds, resolving high-frequency spiking activity and slow varying subthreshold potentials that are challenging to acquire simultaneously at a fixed sampling rate. **d** The PE-CMOS pixel schematic with 6 transistors (T1-T6), a photodiode (PD), and an output (OUT). RST, TX, and SEL are row control signals. EX is a column signal that controls pixel exposure. **e** The pixel layout. The design achieves programmable pixel-wise exposure while maximizing the PD fill factor for high optical sensitivity.

resolution to be the same as the sCMOS at 1.25 ms. Although each PE-CMOS pixel has a low sampling speed of 200 Hz, the PE-CMOS can acquire the ROI at an equivalent of 800 Hz with pixels at different phases. In the subsequent experiments, we used the PE-CMOS Chip B prototype with a 10 μm pixel pitch. It is chosen over Chip A because it has a better fill factor, which should translate to better performance for low-light imaging applications. The pixels exposures are arranged in the tiled configurations by controlling the EX signals using an off-chip FPGA (Xilinx Kintex-7), which also receives the pixel outputs and transmits them to a computer through a PCIe interface, using firmware and software of the Open-Ephys ONIX[16] and Bonsai[17].

We first show that the temporally staggered PE-CMOS sampling configuration achieves high temporal resolution even with a low per-pixel sampling rate. This is seen directly in the PE-CMOS outputs of different phases (Fig. 2c). With a 5 ms exposure and a slow sampling rate of 200 Hz, individual pixels are at a sub-Nyquist rate to sample ASAP3 spikes with 6 ms half-width. This could lead to aliasing, causing the samples to miss the spike (one example is indicated by the black

arrow in Fig. 2c, under phase $\Delta = 0$ ms). Nevertheless, by applying various phase shifts to the neighboring pixels, the spike exceeding the temporal resolution of a single phase was guaranteed to be captured by other phase-shifted pixels (Fig. 2c, black arrow, phase $\Delta = 1.25$ ms). We then linearly interpolated across the ROI and obtained an 800 Hz equivalent time series capturing all the spikes (Fig. 2c). The interpolation process solves a ridge regression, described in detail in the Methods sections. During the interpolation, parameters are chosen to avoid overfitting the noise (Methods). The multi-phasic sampling's ability to resolve aliasing was further recapitulated in a benchtop experiment, where phase-shifted PE-CMOS pixels sampled at 250 Hz accurately captured a fast spike train (8 ms inter-spike interval) at a 1000 Hz temporal resolution (Supplementary Fig. 3, 4, and Methods).

The ability to achieve high temporal resolution with a lower sampling rate allows PE-CMOS to use prolonged pixel exposure to improve the SNR. To demonstrate, we compared the PE-CMOS with the sCMOS outputs over the same ROI (Fig. 2a-f). To ensure a fair comparison of signals from equivalent areas, we evenly divided the

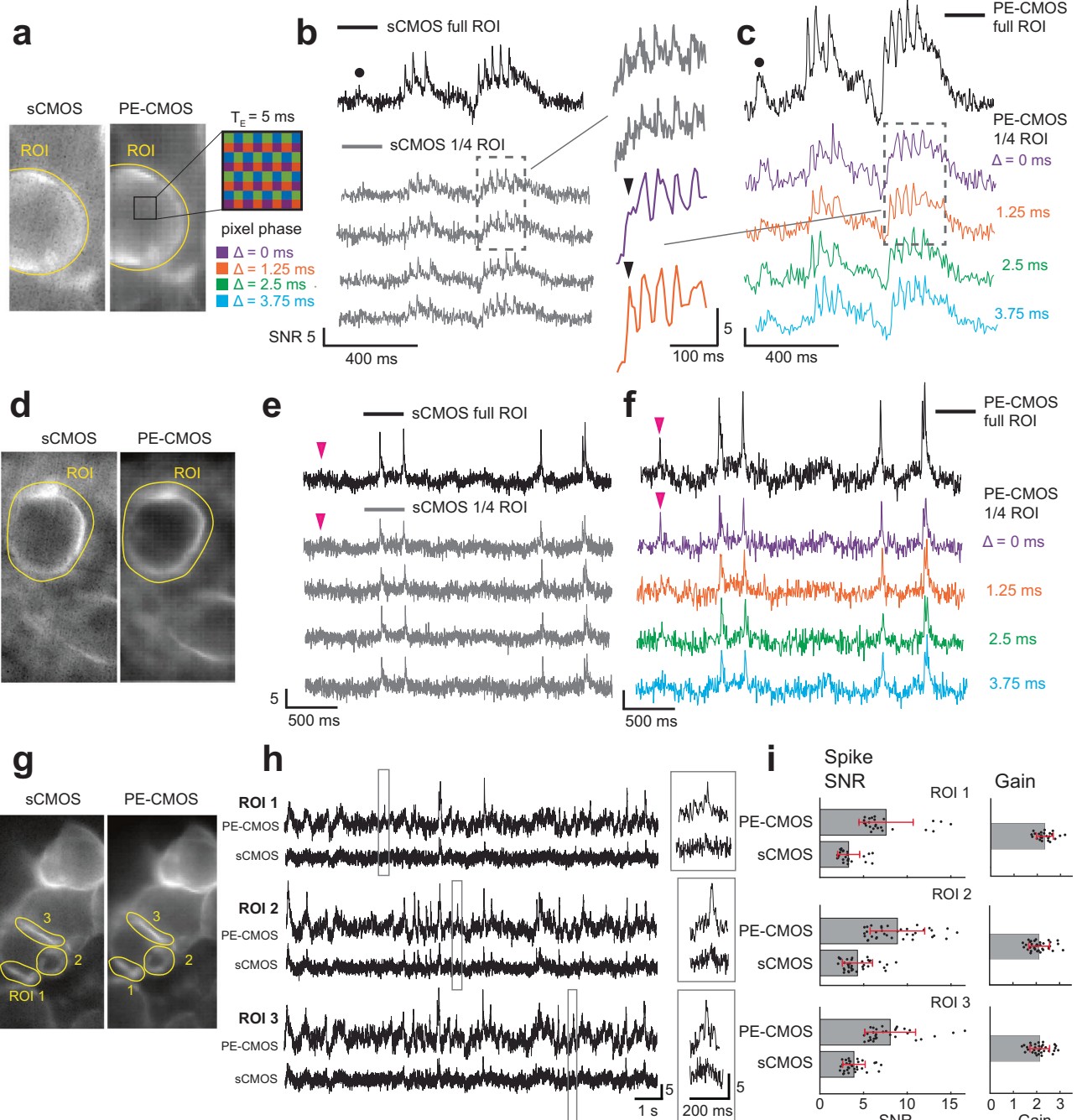

**Fig. 2 | Comparison of the PE-CMOS with an sCMOS camera. a** Maximum intensity projection of the sCMOS (Hamamatsu Orca Flash 4.0 v3) and the PE-CMOS videos of a cultured neuron expressing the ASAP3 GEVI protein. **b** ROI time series from the sCMOS sampled at 800 Hz with pixel exposure ($T_E$) of 1.25 ms. Black trace: ROI time series. Gray trace: the time series each with 1/4 pixels of the ROI. Plotted signals are inverted from raw samples for visualization. **c** simultaneously imaged ROI time series of the PE-CMOS. Colored trace: the time series of phase-shifted pixels at offsets (Δ) of 0, 1.25, 2.5, and 3.75 ms each contain 1/4 pixels of the ROI. All pixels are sampled at 200 Hz with $T_E$ = 5 ms. Black trace: the interpolated ROI time series with 800 Hz equivalent sample rate. Black arrows: An example showing a spike exceeding the temporal resolution of a single phase is captured by phase-shifted pixels. Black circles: an example subthreshold event barely discernable in sCMOS is visible in the pCMOS output. **d**, **e**, **f**: same at panels (**a**, **b**, **c**) with an example showing a spike captured by the PE-CMOS but not resolvable in the sCMOS output due to low SNR (marked by the magenta arrow). **g**, **h** comparison of signal quality from smaller ROIs covering parts of the cell membrane. Gray boxes: zoomed-in view of a few examples of putative spiking events. **i** SNR of putative spikes events from ROIs in panel (**g**). A putative spiking event is recorded when the signals from either output exceed SNR > 5. Data are presented as mean values +/- SD, two-sided Wilcoxon rank-sum test for equal medians, $n$ = 93 events, $p$ = 2.99 × 10$^{-24}$. The gain is calculated as the spike SNR in the PE-CMOS divide by the SNR in the sCMOS. All vertical scales of SNR are 5 in all subfigures.

sCMOS pixel in the ROI into four sections (Fig. 2b, e). We then compared the average time series from each section against the PE-CMOS outputs at different phases (Fig. 2c, f), with each phase covering 1/4 of the total pixels of ROI. We also compared the interpolated PE-CMOS signal over the entire ROI with sCMOS output averaged over the equivalent ROI. The PE-CMOS outperforms the sCMOS with superior SNR under the same temporal resolution. The SNR performance difference is particularly evident in the PE-CMOS ability to detect low-SNR subthreshold potentials and spiking events. One example illustrates this advantage, as a spike is distinctly visible in the PE-CMOS signal

(indicated by the magenta arrow in Fig. 2f), while the same event is indiscernible in the simultaneously recorded sCMOS ROI time series due to a low SNR (Fig. 2e, f). Moreover, the PE-CMOS outputs provide a higher SNR when capturing weak subthreshold activities. An example is marked by the black circle in Fig. 2b, c. Furthermore, the PE-CMOS maintained superior SNR performance over the sCMOS even in smaller ROIs covering parts of the cell membrane (Fig. 2g, h). Many events with sufficient SNR ($>5$) in the PE-CMOS outputs were obscured by the noise floor in the sCMOS outputs (Fig. 2h). To quantify the spike SNR difference between the PE-CMOS and sCMOS, we isolated the putative spiking events at either sensor's output ($>5$ SNR) and compared their SNR (Fig. 2i, two-sided Wilcoxon rank-sum test for equal median, $n = 93$ events, $p = 2.99 \times 10^{-24}$). The putative spike events obtained from the PE-CMOS consistently have higher SNR than that obtained from the sCMOS, with an average SNR gain of a factor of two (Fig. 2i).

To make a direct comparison for capturing somatic voltage at low SNR conditions, we used a patch clamp to measure the intra-cellular potential while performing simultaneous GEVI imaging with both sCMOS and PE-CMOS cameras (Fig. 3). To excite cells during imaging, we injected 200 ms duration current pulses of various amplitudes ($+600$ pA to 0 pA with gradually decreasing 40 pA steps). To quantify the GEVI signal measured by PE-CMOS and sCMOS, we can measure the GEVI response associated with each current injection pulse. The GEVI pulse amplitude is defined as the difference between the average GEVI intensity during each current pulse and the average GEVI intensity 100 ms before and after the pulse (Fig. 3a). GEVI pulse amplitudes are converted into SNR by dividing the noise standard deviation. The noise standard deviation is measured from the GEVI intensity in the absence of current injection pulses. GEVI pulse amplitudes in the PE-CMOS are consistently higher than in sCMOS (Fig. 3a, bar plot). This can be explained by reference to the frequency response of the PE-CMOS (Supplementary Fig. 3e). Due to 4 times longer pixel exposure, the PE-CMOS applies a high amount of gain to a lower frequency signal than sCMOS, which has a uniform gain profile across the frequency range.

However, one could ask, would sampling at 200 Hz with sCMOS achieve the same results? We can mimic an Fs = 200 Hz sCMOS signal by convolving an Fs = 800 Hz signal with a 4 ms box function followed by downsampling by a factor of 4. This would filter out the noise at high frequencies, which increases the SNR of low-frequency GEVI pulse (Fig. 3c, d). However, the resulting signal has a temporal resolution of only 200 Hz, which is insufficient to capture spiking activities, especially at resolving spikes with low inter-spike intervals. We can identify spike positions (Fig. 3c, d, red arrows) with electrophysiology recordings and examine the corresponding GEVI signal amplitude from the PE-CMOS and sCMOS cameras. At Fs = 800 Hz, the sCMOS signal exhibits low SNR, making the spikes less distinguishable from noise than those captured by the PE-CMOS. By filtering high-frequency noise, the sCMOS signal at 200 Hz improves the SNR of some of the spikes. However, this reduced sampling rate leads to aliasing, negatively impacting the amplitudes of other spikes, particularly those with short inter-spike intervals (Fig. 3c, d). The sCMOS output at Fs = 200 Hz has a Nyquist resolution (defined as 2/Fs) of 10 ms. In this case, spikes with inter-spike intervals of 17.4 and 19.5 ms are aliased, causing the spike amplitude to decrease drastically to the point that it can no longer be resolved (Fig. 3d, red arrows). On the other hand, PE-CMOS minimizes the aliasing effect with a 2.5 ms Nyquist resolution, 4 times better than sCMOS at Fs = 200 Hz, preserving the spike amplitude. The ability of PE-CMOS to avoid aliasing is also replicated in Supplementary Fig. 3, with a controlled experiment using an LED to produce optical spike signals spatially uniform across the sensor.

It is also important to note that among the spikes invoked by the current injection pulse, the first spike (shown in Fig. 3d and marked with blue arrows) poses a significant sampling challenge for the image

sensors. This is due to its high-frequency component, which is attributed to the sharp rising edges. Capturing these spikes with sCMOS sensor sampling at 800 Hz leads to low SNR (Fig. 3d). Sampling it at 200 Hz leads to aliasing, which decreases the spike's amplitude (Fig. 3d). In contrast, the PE-CMOS preserves these high-frequency components more effectively than sCMOS (Fig. 3d). This illustrates the PE-CMOS's advantage at capturing high-frequency signals in noisy conditions.

In this experiment, one potential ambiguity might be the GEVI signals at the end of some current pulses (Fig. 3c, black arrow). While intracellular potential shows a flat response, the GEVI signals in both PE-CMOS and sCMOS exhibit significant amplitude variations, which could be mistaken for spiking events. Given that this phenomenon is observed in the outputs from both PE-CMOS and sCMOS sensors (at 800 and 200 Hz), we believe this is not an artifact specific to the PE-CMOS sensor. Instead, it likely results from the response of the GEVI indicators. To ensure the PE-CMOS's interpolation process does not introduce systematic artifacts, we examine the interpolation process in detail (Methods), even when interpolating an ROI with pixels of uncorrelated activity. We showed that interpolating uncorrelated pixels yields an approximation of their average (Supplementary Fig. 8). The selection of regression parameters in the interpolation process minimizes the overfitting of noise.

The superior SNR performance of the PE-CMOS compared to the sCMOS is attributed to the optimally chosen pixel exposure. We modeled the relationship between pixel exposure and SNR using the known GEVI time constant of action potentials (Supplementary Fig. 5a and Methods). The pixel output is then computed as the area under the GEVI spike, and the major noise sources are photon shot noise and circuit readout noise (Supplementary Fig. 5a and Methods). Pixel SNR increases with exposure, attributed to two reasons: first, the photon shot noise is reduced relative to the signal as more photons are integrated by the pixel, and second, the readout noise decreases due to a slower sampling rate (Supplementary Fig. 5b and Methods). The most optimal exposure to maximize SNR is around 4–8 ms, roughly matching the width of a GEVI spike. Pixel exposure extending longer than the spike width has a diminishing benefit on the SNR. To sample ASAP3 spikes, this model predicted that increasing exposure from 1.25 to 5 ms corresponds to ~2–3 fold of spike SNR improvement depending on the pixel read noise, baseline fluorescence level, and the fluorescence signal dF/F (Supplementary Fig. 5c). The model prediction matched our empirical measurements (Fig. 2i), where the PE-CMOS ($T_E = 5$ ms) results in a spike SNR gain of greater than two compared to the sCMOS ($T_E = 1.25$ ms). This is a significant signal quality improvement without increasing excitation power and allows for extracting meaningful physiological signals that conventional sCMOS cameras cannot. It is also worth noting that due to longer pixel exposure, time-staggered sampling using PE-CMOS outperformed sCMOS, despite its slightly worse read noise performance and lower QE (readnoise:2.67 e⁻ at room temperature, with QE of 68% without microlens), compared to that of the sCMOS (datasheet readnoise:1.6 e⁻ at −16 deg. C, with QE of 82%).

## Sampling voltage signals at multiple temporal resolutions with spatially varying pixel exposures

The PE-CMOS can also be used to optimize pixel sampling patterns based on temporal features of the ROI. As we demonstrated, the optimal exposure to maximize SNR depends on the signal's shape (Supplementary Fig. 5). However, an ROI may contain events of diverse temporal characteristics. For example, a neuron may exhibit spiking events and subthreshold activities of different intensities and frequencies. In such cases, using a fixed pixel exposure cannot ensure the optimal detection of all signals representing important and distinct physiological mechanisms. A better strategy is to sample the

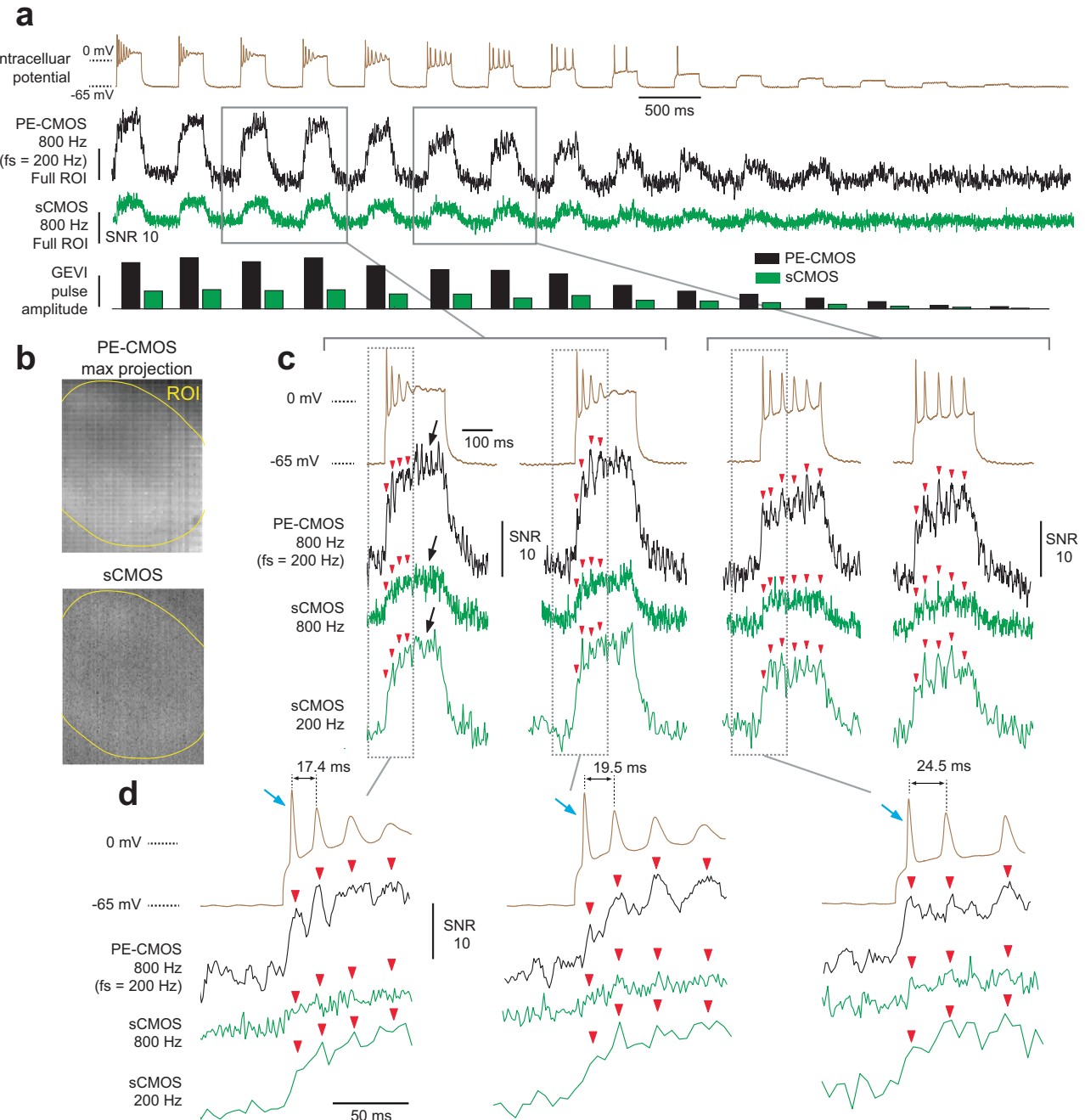

**Fig. 3 | imaging of a patched cell using both the PE-CMOS and sCMOS sensor.**
**a** The intracellular potential of the cell and the ROI GEVI time-series of the PE-CMOS and sCMOS. GEVI pulse amplitude is the change in GEVI signal corresponding to each current injection pulse. It is measured as the difference between the average GEVI intensity during each current pulse and the average GEVI intensity 100 ms before and after the current injection pulse. GEVI pulse amplitude is converted into SNR by dividing the noise standard deviation. **b** max. projection of the cell in PE-CMOS and sCMOS. **c** zoomed in view of the intracellular voltage and GEVI pulses in (**a**). The red arrow indicates spike locations identified from the intracellular voltage.

The black arrows indicate a time where intracellular potential shows a flat response when the GEVI signals in both PE-CMOS and sCMOS exhibit significant amplitude variations. These can be mistaken for spiking events. **d** zoomed in view of (**c**) showing the PE-CMOS trace can resolve two spikes with small inter-spike interval, while sCMOS at 800 Hz and 200 Hz both fail to do so. The blue arrows point to the first spike invoked by the current pulse. While the sharp rising edges make them especially challenging for image sensors to sample, the PE-CMOS can preserve their amplitudes better the sCMOS.

pixels of the ROI at different exposures and speeds. To illustrate this, we configured the PE-CMOS to sample the ASAP3 activity of a cell soma with pixel exposure ranging from 15.4 ms down to 1.9 ms (Fig. 4a, b). This configuration samples an ROI at 64–520 Hz, aiming to simultaneously capture fast spike bursts, low SNR spikes, and subthreshold activities. The ROI time series showed that fast pixels (520 Hz) can separate high-SNR burst events in time but cannot

reliably detect slower and weaker GEVI activities due to low SNR (Fig. 4d–f). On the other hand, pixels sampling at moderate (260 Hz) and slow (<130 Hz) speeds improved the SNR for low-frequency signals and detected low SNR candidate events for subthreshold activities or low SNR spikes. Combining pixels at different speeds, we reconstructed a 1040 Hz equivalent ROI time series through interpolation (Methods), capturing high-SNR spike bursts, low SNR

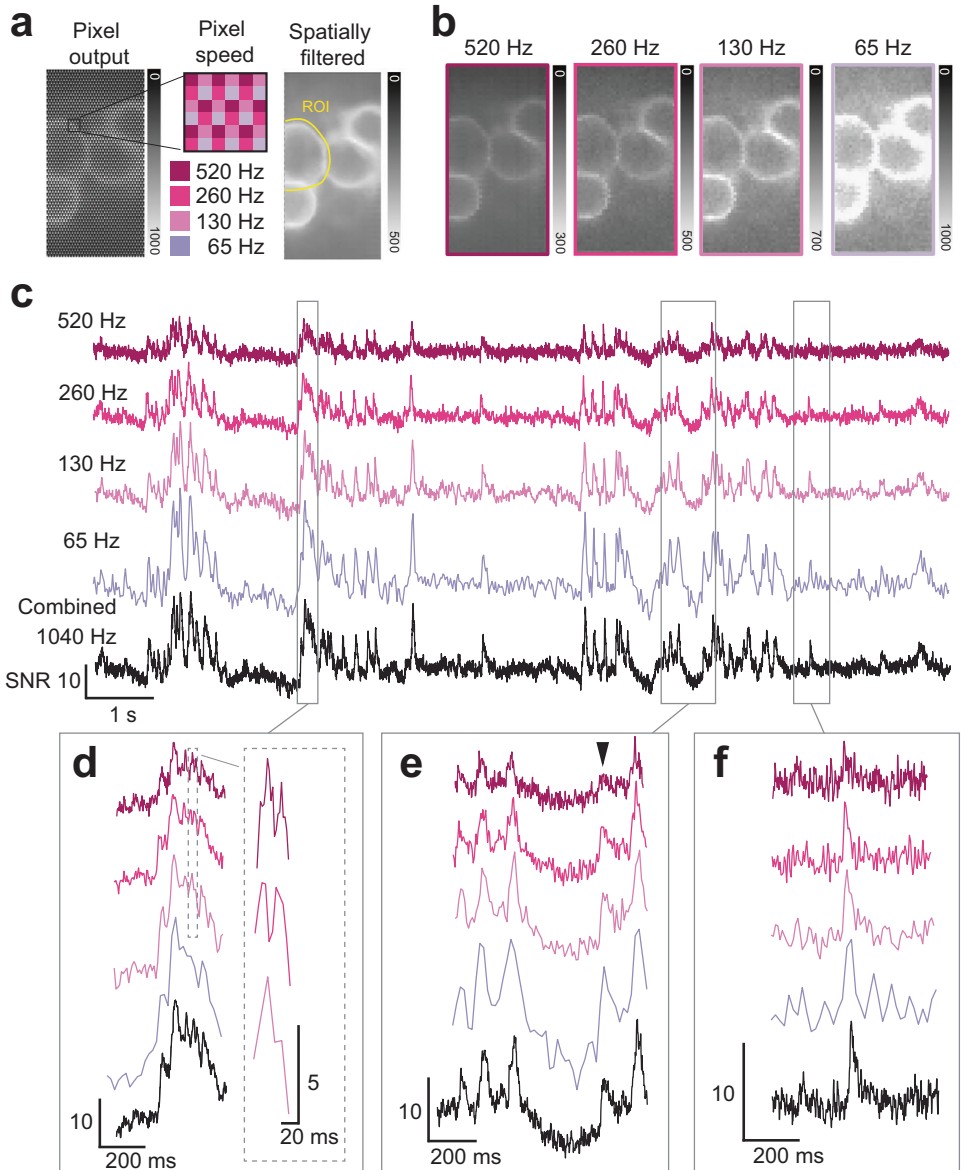

**Fig. 4 | Multi-exposure/speed sampling to simultaneously maximize SNR of spiking events and subthreshold activity. a** Maximum intensity projection of the PE-CMOS videos, raw and filtered (2 × 2 spatial box filter) output at full spatial resolution. Intensity is measured by digital bits (range: 0–1023). **b** Maximum intensity projection divided into four sub-frames according to pixel sampling speed, each with 1/4 spatial resolution. **c** The ROI time series from pixels of different speeds (colored trace). Black trace: a 1040 Hz equivalent signal interpolated across all ROI pixels. **d** Fast sampling pixels (520 Hz) resolves high-SNR spike bursts. **e**–**f** Pixels with more prolonged exposure ($T_E$ = 2.8–5.7 ms) improves SNR to detect weak subthreshold activity (black arrow) and (**f**) low SNR spike. The vertical scale of SNR is 10 unless otherwise noted.

spikes, and weak slow-varying subthreshold activities using a 260 Hz equivalent sensor readout speed (Fig. 4c–f).

Sampling with spatially varying pixel exposures also enhances the dynamic range for the FOV with large brightness variation, often caused by uneven fluorescence indicator expression. To sample these scenes, uniform pixel exposure can cause signal saturation in bright areas and low SNR in dim ones. Unlike in two-photon microscopy, adjusting excitation power per pixel[18] is challenging in wide-field excitation without complex optical setups. Spatially varying pixel exposures capture a 2 × 2 pixel region with different exposures (Fig. 4b), simultaneously capturing bright areas with short exposures to prevent oversaturation and enhancing dim area's SNR with longer exposures. The pixel exposure can also be adaptively adjusted in a closed-loop system to optimize the dynamic range further, similar to our previously demonstrated method[19].

## Discussion

### Application beyond fluorescence imaging

The advantages of programmable pixel-wise exposure extend beyond fluorescence microscopy to general imaging applications, particularly in capturing scenes with a high dynamic range of motion and light intensity. For example, the time-staggered pixel pattern can track high-speed motion in low-light conditions with long exposures. For instance, the PE-CMOS sensor with 25 Hz sample rate resolves a motion of 100 frame/s (Supplementary Fig. 6).

### Future applications

The PE-CMOS sensor enables several promising future applications. Its CMOS-based design offers scalability to higher pixel numbers and speed to meet the demand of various heterogenous imaging applications. For instance, the PE-CMOS allows configurations of pixel

exposures to optimally match the temporal characteristics of dendritic potentials evoked by different neurotransmitters[20] at various synapses. Additionally, using pixels sampled at different speeds, the PE-CMOS may enable us to simultaneously image multiple fluorescent indicators with different intensity ranges and time constants, such as calcium and GEVI signals[21]. Finally, the PE-CMOS power efficiency, resulting from its ability to achieve high temporal resolution with low sampling speed, makes it suitable for miniaturized microscopes[22–25] to enable kHz GEVI imaging from freely moving rodents. Under a sensor speed of 100 to 250 Hz, which is typically limited to capturing slow fluorescent dynamics like calcium indicators, the PE-CMOS has the potential to enable GEVI imaging at 400-1000 Hz. Ultimately, this will provide an opportunity to directly image in vivo neural voltage activity to investigate fast-spiking activities[26] underlying complex behavior.

## Potential limitations

We have demonstrated the performance of the PE-CMOS through in vitro experiments. We foresee several possible limitations that would benefit from in vivo validations. First, during in vivo imaging in freely behavioral mice, the cells can experience small movements in the FOV. To correct this movement, we can track each cell's position at each frame and re-align them through transformation. Accurate motion tracking depends on (1) temporal resolution: how accurately we can sample the cell motion, and (2) motion blurring: how accurately we can isolate the exact position of the cell. The PE-CMOS can achieve higher temporal resolution, but the blurring is determined by its pixel exposure duration. Blurring will only become a problem for high-speed cell movement. Although we do not know the exact amount of cell movement, previous calcium imaging in behaving mice offers estimations. These microscopes sample the FOV at 10–30 Hz (with 100 ms to 30 ms of blurring) and have not reported blurring-induced inaccuracy in motion correction. Therefore, we do not expect this will be a problem for GEVI imaging, where the PE-CMOS pixel sampling rate is >200 Hz with exposure < 4 ms. However, this claim needs careful validation from in vivo experiments.

Second, phase-shifted pixels enhance the bandwidth of the sampled ROI time series, but it does not eliminate the narrowband attenuations (e.g. seen at 250 and 500 Hz in Supplementary Fig. 3e) induced by long pixel exposures. While these attenuations may not affect the detection of spikes with a broad frequency spectrum, they may pose a problem when the ROI contains specific signals of interest (such as brain oscillations) that fall into these frequency ranges. To resolve this, the PE-CMOS can employ pixels with varied exposure durations within the ROI (Supplementary Fig. 4). These pixels sample the ROI at different speeds with varying exposure. While certain frequencies may be attenuated by pixels with a specific exposure duration, they are preserved by other pixels with different exposure lengths. Consequently, the frequency spectrum of the ROI displays no narrowband attenuation.

## Future improvements

The PE-CMOS sensor can be further improved for better sensitivity and noise performance. Without refabricating the chip, we could implement a layer of microlens array on top of the pixel array to enhance the pixel quantum efficiency, a widely used process for image sensors. On the system side, we could implement active cooling to reduce the chip temperature to achieve lower read noise. The demonstrated PE-CMOS circuits architecture is a generalized circuit design, which could be implemented using the advanced CMOS image sensor fabrication processes (such as back-illuminated photodiode processes) to further take advantage of enhanced photodiode sensitivity and low noise transistors to improve the overall optical performance.

## Methods

### Finding the optimal exposure to capture GEVI spikes

The exposure duration ($T_E$) of a pixel has a direct impact on its SNR, and the most optimal $T_E$ for maximizing the SNR depends on the shape of the signal. We created a model to demonstrate how the SNR and $T_E$ are correlated when sampling a GEVI spike. By using this model, we can select an optimal $T_E$ that maximizes the SNR.

We first model the shape of the GEVI signal, $\mathbf{v}(t)$, during a spiking event, which is approximated as a combination of double exponential functions[2–4]:

$$\mathbf{v}(t) = \begin{cases} A_R e^{t/\tau_{Rf}} + B_R e^{t/\tau_{Rs}} & \text{when } t < 0 \\ A_F e^{-t/\tau_{Ff}} + B_F e^{-t/\tau_{Fs}} & \text{when } t \geq 0 \end{cases} \quad (1)$$

where $\tau_{Rf}$ and $\tau_{Rs}$ characterize the spike rise times at fast and slow timescales, respectively; and $\tau_{Ff}$ and $\tau_{Fs}$ are the respective fall time constants. $A$ and $B$ are two scaling constants of the GEVI indicators. Supplementary Fig. 5a plots the shape of $\mathbf{v}(t)$ with parameters of three different GEVI indicators (ASAP3[3], Archon[4], and Voltron[2]), with the spike peak normalized to 10% dF/F.

When a pixel samples $\mathbf{v}(t)$ with an exposure duration of $T_E$, the resulting output, $\mathbf{y}[n]$, is obtained as discrete samples of the convolution between $\mathbf{v}(t)$ and the exposure, $\mathbf{e}(t)$:

$$\mathbf{y}(t) = \mathbf{e}(t) * \mathbf{v}(t), \mathbf{e}(t) = \begin{cases} 1 & 0 \leq t \leq T_E \\ 0 & otherwise \end{cases} \quad (2)$$

$$\mathbf{y}[n] = \mathbf{y}(nT_E) + \sigma[n] \quad (3)$$

where $\sigma[n]$ is the sampled noise containing both shot noise and circuit read noise. We assume that the shot noise is described by a Poisson distribution with a rate factor $\lambda$ equal to the signal at the pixel. As such, its power is expressed as:

$$\mathbf{N}_{shot} = \mathbf{y}[n] + F \cdot T_E$$

where $F$ denotes the baseline fluorescent signal, represented here as a constant for simplicity.

The read noise of a circuit comprises two components: flicker noise and thermal noise. In pixel design, the correlated double sampling technique can effectively eliminate the flicker noise, which is more prominent at low frequencies due to its 1/f noise density profile. Therefore, we can estimate the circuit read noise using only thermal noise that has a white power spectrum over the circuit bandwidth, with the power given by:

$$\mathbf{N}_{read} \approx \mathbf{N}_{thermal} = N_0 / T_E$$

where $N_0$ is a noise density measured in V/Hz, and $1/T_E$ is the circuit bandwidth. The magnitude of thermal noise depends on the circuit bandwidth, which is inversely proportional to the exposure time of the pixel.

We can now derive an analytic expression of the peak pixel SNR with respect to $T_E$:

$$SNR = \sqrt{\frac{S}{N_{shot} + N_{read}}} = \sqrt{\frac{\max\{\mathbf{y}\}^2}{N_0/T_E + \max\{\mathbf{y}\} + F \cdot T_E}} \quad (4)$$

$$\approx \frac{\max\{\mathbf{y}\}}{\sqrt{N_0/T_E + F \cdot T_E}} \text{ when } F \gg \max\{\mathbf{y}\}$$

Here, the SNR is defined as the ratio between signal amplitude and the root-mean-squared noise. We plotted Eq. 4 for various GEVI indicators

for exposure time $T_E$ (Supplementary Fig. 5b, parameters: F = 20$e^-$, peak dF/F = 10%, $N_0$ = 2.6 $e^-_{rms}$). For comparison across different GEVI indicators, we normalize the SNR by dividing it by the peak value. For all the GEVI indicators, the SNR improves rapidly with longer $T_E$ but drops gradually as extra integration time adds more shot noise than signal power due to the baseline fluorescent. To maximize the SNR for GEVI spikes, the most optimal $T_E$ is 4–8 ms, approximately the half-width of the action potential.

## Using phase offset pixel-wise exposures to increase temporal resolution

Maximizing the SNR requires a pixel exposure duration approximately equal to the spike width (Supplementary Fig. 5). However, this can limit the pixel's temporal resolution and potentially create ambiguity in detecting spiking events. To demonstrate this, we conducted a benchtop experiment using a LED[27] light source to simulate a spike burst with each spike having a 4 ms half-width and an 8 ms interspike interval (Supplementary Fig. 3a). To accurately detect this spiking event, it is necessary to have enough time samples to capture the spike shape and interspike interval. At a minimum, we need three temporal samples to capture a spike event (one sample at the peak and one on the trough on either side of the peak) or an interspike interval. However, prioritizing SNR maximization using a pixel exposure duration of 4 ms will not provide enough samples to detect the spikes in this example and their separations unequivocally. Sampling this signal at 250 Hz may result in aliasing, making it impossible to distinguish between the spikes (Supplementary Fig. 3b).

Introducing phase shifts between pixels in PE-CMOS enables us to enhance the temporal resolution of an ROI without increasing the sampling rate of individual pixels. For example, using four pixels, we can sample the ROI time-series at four phases (Supplementary Fig. 3c). This approach helps to overcome aliasing issues that may arise when sampling the ROI without pixel phase offsets. Spikes that are not accurately detected at one phase can be captured at phase-shifted pixels (Supplementary Fig. 3d). Interpolating across these pixels allows us to reconstruct spiking events at 1000 Hz resolution, with eight samples separating the spike peaks (Supplementary Fig. 3f).

In this example, the phase-shifted sampling pattern enables us to increase the bandwidth of the ROI by a factor of four, without the need to increase the sampling speed of each individual pixel. At a sampling rate of 250 Hz, a single pixel's bandwidth is limited to 125 Hz. Frequencies above 125 Hz fold into the pixel's bandwidth and cause aliasing (Supplementary Fig. 3e), which makes it difficult to distinguish between frequency components. By introducing phase-offset pixels, the combined outputs of four pixels sample the ROI at a much higher frequency, effectively extending the bandwidth to 500 Hz, four times that of the individual pixel, and avoiding the aliasing effects that occur with single pixels (Supplementary Fig. 3e).

Using phase-shifted pixels enhances the bandwidth of the sampled ROI time series. However, it does not eliminate the narrowband attenuations (seen at 250 and 500 Hz in Supplementary Fig. 3e), since these are inherent to the exposure function, $\mathbf{e}(t)$. While these attenuations may not affect the detection of spikes with a broad frequency spectrum, they may pose a problem when the ROI contains specific signals of interest that fall into these frequency ranges. The PE-CMOS can employ pixels with varied exposure durations within the ROI (Supplementary Fig. 4). These pixels sample the ROI at different speeds with varying $\mathbf{e}(t)$. While certain frequencies may be attenuated by pixels with a specific $\mathbf{e}(t)$, they are preserved by other pixels with different exposure lengths. Consequently, the frequency spectrum of the ROI displays no narrowband attenuation (Supplementary Fig. 4).

## Theoretical analysis on using multi-phase sampling to eliminate aliasing

In an ROI with highly correlated pixel outputs, we demonstrate that combining the outputs from low-frequency pixels, each sampled at distinct phase-shifted exposures, can eliminate aliasing and improve the temporal resolution of the ROI average time series.

We assume a light impulse, $\boldsymbol{\delta}(t)$, is simultaneously sampled by 4 pixels (Supplementary Fig. 7a), $y_1, ..., y_4$ with an exposure function $\mathbf{e}(t)$:

$$\mathbf{y}_k(t) = \mathbf{e}(t)^* \boldsymbol{\delta}(t), \mathbf{e}(t) = \left\{ \begin{array}{ll} 1 & 0 \le t \le T_E \\ 0 & otherwise \end{array} \right\}, k \in \{1,4\},$$

$\mathbf{y}_k(t)$ is then sampled by an ADC at a period of $T_E$. $T_E$ is also the exposure duration. We then write the discretized version of $\mathbf{y}_k(t)$:

$$\mathbf{y}_k[n] = \mathbf{y}_k(t) \cdot \sum_{n=-\infty}^{\infty} \boldsymbol{\delta}(t - nT_E)$$

with frequency spectrum:

$$\mathbf{Y}_k(f) = \frac{1}{T_E} \sum_{n=-\infty}^{\infty} \mathbf{E}\left(f - \frac{n}{T_E}\right)$$

Where $\mathbf{E}(f - \frac{n}{T_E})$ are the replicas of the exposure function's spectrum resulting from sampling. We can plot $\mathbf{Y}_k(f)$ to see that the majority of the frequency spectrum is aliased, resulting in signal distortion (Supplementary Fig. 7b).

By relying on the phase difference between neighboring pixels, we can eliminate the aliasing effect without increasing each pixel's sampling speed. To achieve this, relative to the phase of pixel 1, we can introduce phase shifts of $\frac{T_E}{4}, \frac{T_E}{2}, \frac{3T_E}{4}$ to pixels 2, 3, and 4, respectively (Supplementary Fig. 7c). These pixels' spectrum, $\mathbf{Y}_k(f)$, becomes:

$$\mathbf{Y}_k(f) = \frac{1}{T_E} \sum_{m=-\infty}^{\infty} \mathbf{E}\left(f - \frac{n}{T_E}\right) e^{-j2\pi\frac{n(k-1)}{4}}$$

where $k \in \{1,4\}$. If we average these four pixels, the resulting spectrum becomes:

$$\mathbf{Y}_{avg}(f) = \frac{1}{4}\sum_{k=1}^{4} \mathbf{Y}_k(f) = \frac{1}{4T_E} \sum_{m=-\infty}^{\infty} \mathbf{E}\left(f - \frac{n}{T_E}\right)\left(1 + e^{-j2\pi\frac{n}{4}} + e^{-j2\pi\frac{n}{2}} + e^{-j2\pi\frac{3n}{4}}\right)$$

since $\mathbf{Y}_{avg}(f) = 0$ when $m$ is not a multiple of 4, the above equation can be rewritten as:

$$\mathbf{Y}_{avg}(f) = \frac{1}{4T_E} \sum_{n=-\infty}^{\infty} \mathbf{E}\left(f - \frac{4n}{T_E}\right)$$

which spaces the frequency replicas of $\mathbf{E}(f)$ away from each other and avoids aliasing (Supplementary Fig. 7d). The average spectrum of phase-shifted pixels increases the Nyquist bandwidth of individual pixels by four times, without an increase in per-pixel sampling rate. From a time-domain viewpoint, the combined phase-shifted pixel outputs are equivalent to the sampling of the signal, $\mathbf{v}(t)$, at a higher rate of $4/T_E$.

## ROI signal interpolation

We employ linear interpolation to compute the ROI time series from the pixel outputs. For an ROI containing $N$ pixels, let $\mathbf{v}_n, n = \{1 \dots N\}$, of discrete length $L$ represent the fluorescence signal at $n$ th pixel. The sampled value at this pixel, $\mathbf{y}_n$, of discrete length, $M$ is:

$$\mathbf{y}_n = \mathbf{E}_n \mathbf{v}_n \tag{5}$$

where $\mathbf{E}_n \in \mathbb{R}^{M \times L}$ denotes the sampling matrix representing pixel exposure and sampling operation that converts the fluorescent signal $\mathbf{v}_n \in \mathbb{R}^L$ into the pixel outputs, $\mathbf{y}_n \in \mathbb{R}^M$.

Now, we can define an arbitrary signal, $\mathbf{v}_{ROI}$, and rewrite each $\mathbf{y}_n$ as:

$$\mathbf{y}_n = \mathbf{E}_n \mathbf{v}_{ROI} + \mathbf{E}_n(\mathbf{v}_n - \mathbf{v}_{ROI})$$

Note that we do not make the assumption that $\boldsymbol{v}_{ROI}$ is the average of the pixels.

if we write the difference term $\boldsymbol{\varphi}_n = \mathbf{E}_n(\mathbf{v}_n - \mathbf{v}_{ROI})$, we can then concatenate the output of all of the $N$ pixels into one vector:

$$\mathbf{y} = \mathbf{E}\mathbf{v}_{ROI} + \boldsymbol{\varphi} \qquad (6)$$

where

$$\mathbf{y} = \begin{bmatrix} \mathbf{y_1} \\ \vdots \\ \mathbf{y_N} \end{bmatrix}, \mathbf{E} = \begin{bmatrix} \mathbf{E_1} \\ \vdots \\ \mathbf{E_N} \end{bmatrix} \text{ and } \boldsymbol{\varphi} = \begin{bmatrix} \boldsymbol{\varphi_1} \\ \vdots \\ \boldsymbol{\varphi_N} \end{bmatrix}$$

and

Given Eq. 6, we can find $\mathbf{v}_{ROI}$ using a ridge regression by enforcing a $L_2$ norm penalty:

$$\mathbf{v}_{ROI} = \min_{\mathbf{v}_{ROI}} \frac{1}{2} \left\| \mathbf{y} - \mathbf{E}\mathbf{v}_{ROI} \right\|_2^2 + \lambda_{ridge} \left\| \mathbf{v}_{ROI} \right\|_2^2$$

and further derive its closed-form solution:

$$\hat{\boldsymbol{v}}_{ROI} = \left( \mathbf{E}^{\mathrm{T}}\mathbf{E} + \lambda_{ridge}\mathbf{I} \right)^{-1} \mathbf{E}^T \mathbf{y} \qquad (7)$$

where $\hat{\boldsymbol{v}}_{ROI}$ is the least-squared estimate of $\boldsymbol{v}_{ROI}$, and the hyperparameter, $\lambda_{ridge}$, controls the trade-off between the $L_2$-norm of $\mathbf{v}_{ROI}$ and reconstruction error, denoted by $\|\mathbf{y} - \mathbf{E}\mathbf{v}_{ROI}\|_2 = \|\boldsymbol{\varphi}\|_2$.

Solving this regression should push $\mathbf{v}_{ROI}$ to be close to the average of the pixels ($\mathbf{v}_{ROI}^- = \frac{1}{N}\sum_{n=1}^N \mathbf{v}_n$), which by definition, minimizes the L2 norm of $\|\mathbf{y} - \mathbf{E}\mathbf{v}_{ROI}\|_2$. However, in the presence of noise, setting $\lambda_{ridge}$ could have a risk of overfitting. To avoid this, in the interpolation method used in our manuscript, we set $\lambda_{ridge}$ to be large, which conservatively underfit our approximation for $\boldsymbol{v}_{ROI}$.

With a large $\lambda_{ridge}$, the term $\left( \mathbf{E}^{\mathrm{T}}\mathbf{E} + \lambda_{ridge}\mathbf{I} \right)^{-1}$ will approximate a scaled identity matrix, and Eq. 7 becomes

$$\hat{\mathrm{v}}_{ROI} \approx \mathrm{E}^T \mathrm{y} \qquad (8)$$

which is equal to up-sampling and interpolation of the pixel outputs $\mathbf{y}$.

We can reinforce our analysis with a simple example in Supplementary Fig. 8. Here, the time series of 4 pixels, $\mathbf{v}_1 \ldots \mathbf{v}_4$, contains uncorrelated signal, with $\boldsymbol{v}_{ROI}$ representing the average of these signals. We mimic the exposure and phase-shifted sampling of these pixels to get $\mathbf{y}_1 \ldots \mathbf{y}_4$. We do this by convolving phase-shifted version of $\mathbf{v}_1 \ldots \mathbf{v}_4$ with a box function of length 4, followed by 4 x down-sampling. We can see that down-sampling aliases the high-frequency part of the signal of $\mathbf{y}_2$, especially at the peak of the spike (Supplementary Fig. 8 black arrow).

We can apply our interpolation outlined in Eq. 8, which results in $\hat{\mathbf{v}}_{ROI}$. We can see that $\hat{\mathbf{v}}_{ROI}$ is proportional to $\boldsymbol{v}_{ROI}$, but underfits the spike at the location pointed by the black arrow in Supplementary Fig. 8. This is because we only sampled $\mathbf{y}_2$ at a single phase since $\mathbf{y}_2$ is uncorrelated with other pixels.

This example shows that even with uncorrelated pixels, interpolating them yields an approximation of the ROI average. As shown, the selection of large $\lambda_{ridge}$ in the ridge regression minimizes the overfitting of noise during interpolation.

## Voltage imaging in dissociated neuron cultures

**Neuron culture preparation and AAV transduction.** All procedures involving animals were performed in accordance with National Institute of Health Guide for Laboratory Animals and approved by the Massachusetts Institute of Technology Committee on Animal. Dissociated hippocampal neurons were prepared from postnatal day 0 or 1 Swiss Webster mice (Taconic) without regard to sex following the protocol[28]. Dissected hippocampal tissue was digested with 50 units of papain (Worthington Biochem) for 6–8 min, and the digestion was stopped with ovomucoid trypsin inhibitor (Worthington Biochem). Cells were plated at a density of 40,000–60,000 per glass coverslip coated with Matrigel (BD Biosciences). Neurons were seeded in 200 μl plating medium containing MEM (Life Technologies), glucose (33 mM, Sigma), transferrin (0.01%, Sigma), Hepes (10 mM, Sigma), Glutagro (2 mM, Corning), Insulin (0.13%, Millipore), B27 supplement (2%, Gibco), and heat inactivated FBS (7.5%, Corning). After cell adhesion, additional plating medium was added. AraC (0.002 mM, Sigma) was added when glia density was 50–70% of confluence. Neurons were grown at 37 °C and 5% $CO_2$ in a humidified atmosphere.

We transduce cultured neurons at 5–7 days in vitro (DIV) by administering ~$10^{10}$ viral particles of AAV9-hSyn-ASAP3 (Janelia Viral Tools) per well (of 24-well plate). Voltage imaging was performed 7–14 days after transduction.

**Microscope setup (without patch clamp).** Cultured hippocampal neurons expressing ASAP3 were imaged on a customized upright fluorescent microscope with a 20 × 1.0NA objective lens (Olympus). The light from a 470 nm LED (Thorlabs) was cleaned with a 469/35 nm band pass filter (Semrock) for excitation. A 488 nm long pass dichroic mirror and a 496 nm long pass filter were used for illumination and emission. Using a 50/50 beam splitter (Thorlabs), the sample image was evenly split onto both the Hamamatsu sCMOS camera and our PE-CMOS sensor for side-by-side comparison.

**Whole cell patch clamp and imaging.** Intracelluar recordings were acquired using the multiclamp 700B amplifier and pCLAMP 10.0 software (Molecular Devices), filtered at 2 kHz and digitized at 10 kHz. Cultured neurons were patched with pipettes filled with a potassium gluconate-based intracellular solution containing the following (in mm): 135 mM K-gluconate, 0.1 mM CaCl2, 0.6 mM MgCl2, 1 mM EGTA, 10 mM HEPES, 4 mM Mg-ATP, 0.4 mM Na-GTP, and 4 mM NaCl, with dropwise addition of 5 M KOH to adjust the pH to 7.2 and addition of potassium gluconate in increments of 25 mg until the final osmolarity reached 290–295 mosmol/kg H2O. The open pipette resistance in these experiments was 3–6 mΩ. The current step protocol consisted of 200-ms-long constant level current steps from + 600 pA–0 pA with gradually decreasing 40 pA steps. During patch clamp, the ASAP3 expressing neuron are simultaneously imaged through a 40 x / 0.6 NA objective lens. Using a 50/50 beam splitter (Thorlabs), the sample image was evenly split onto both the Hamamatsu sCMOS camera and our PE-CMOS sensor for side-by-side comparison.

**ROI time series extraction.** We hand selected the ROI using the max projection image of the PE-CMOS and sCMOS videos. Due to differences in pixel size (PE-CMOS: 10 μm pixel pitch, sCMOS: 6.5 μm), sCMOS ROI contains ~2.3 x more pixels than that of the PE-CMOS ROI. PE-CMOS pixels are arranged in the tile configuration (Supplementary Fig. 2).

For the sCMOS, we computed the ROI time series by averaging all the pixels values. We then filtered the ROI time-series using a band-

pass filter to remove photo-bleaching effects and high-frequency content too close to the sampling Nyquist frequency (Filter parameters: 4th-order IIR, with high-pass cut-off at 0.5 Hz and low-pass cut-off at 360 Hz, 90% of the Nyquist bandwidth, 400 Hz, half of pixel sampling speed). For visualization purpose, we invert the time series as ASAP3 GEVI has negative going action potential waveforms.

For the PE-CMOS, we first computed sub-ROI time series for pixels with the same $T_E$ and $\Delta$. We then filtered each sub-ROI time series using a band-pass filter to remove photo-bleaching effects and high-frequency content too close to the sampling Nyquist frequency. The filter parameters used in Fig. 2, 3 are: 4th-order IIR, with high-pass cut-off at 0.5 Hz and low-pass cut-off at 90 Hz, 90% of the pixel Nyquist bandwidth 100 Hz. The same filter is used in Fig. 4, but with low-pass cut-off changed to 234 Hz, 117 Hz, 58 Hz, and 29 Hz, corresponding to different pixel Nyquist bandwidth 260, 130, 65 and 33 Hz. The ROI time series representing the ROI average are interpolated from sub-ROI time-series using method described earlier. No additional filters are used for the PE-CMOS ROI time-series.

To compare the sCMOS output to that of the PE-CMOS at different $\Delta$ in Fig. 2, which compose of ~1/4 pixels of the ROI, we computed four sCMOS sub-ROI time series each using only 1/4 ROI pixels extracted through spatial downsampling. The average of these 4 sub-ROI time series is equal to the ROI time series.

**Noise estimation.** To estimate the noise of the times series, we find a section of the signal with at least 2 seconds in duration that does not contain any spikes. We then high-pass filter this trace to remove slow subthreshold oscillations (Filter parameters: 4th-order IIR, $f_{cutoff}$ = 50 Hz). The root-mean squared (rms) noise is then calculated as the standard deviation of the filtered time-series. This value is referred to as 1 SNR in the figures and text.

**Putative spike detection.** ROI times-series are first high-pass filtered to remove high amplitude oscillations at low frequency (Filter parameters: 4th IIR filter, $f_{cutoff}$ = 2 Hz). From the filtered PE-CMOS time series, we detected events with SNR > 5 and record their SNR and time of occurrence. We then record the highest SNR in the sCMOS times series within the corresponding time window. We set a 30 ms window centering at the peak time of PE-CMOS to avoid mismatching the peak SNR between PE-CMOS and sCMOS time series.

### General imaging applications using PE-CMOS
We demonstrated two general imaging examples that benefits from using the PE-CMOS. In the first example, the PE-CMOS pixels are configured to have fixed exposure with $T_E$ = 40 ms and $\Delta$ = 0, 10, 20, 30 ms (Supplementary Fig. 6a) This corresponds to pixel sampling speed of 25 Hz. We imaged the motion of bouncing ball with static background (Supplementary Fig. 6b). Staggered exposure embeds the motion within full resolution frames (Supplementary Fig. 6b). We split each frame into four subframe organized according to pixel phase, which yields a 100 Hz equivalent video at 1/4 spatial resolution. From the high-speed video, we can track the motion of the bouncing ball with temporal precision of 10 ms (Supplementary Fig. 6d), which is not possible using a conventional camera where pixels are uniformly sampled at 25 Hz.

### Reporting summary
Further information on research design is available in the Nature Portfolio Reporting Summary linked to this article.

## Data availability
The raw data generated in this study have been deposited in the Zenodo database under accession code https://zenodo.org/records/10826791. The PE-CMOS image sensor prototypes are available upon request. We will provide all the information required to replicate the design. However, the design documents use proprietary circuit models and files from the commercial foundry that are protected under the NDA. We can share the design with the interested party after they have secured the same NDA with XFAB. To interface with the chip, we used open-source hardware, firmware and software API from the Open-Ephys ONIX[16] project and Bonsai[16].

## Code availability
The script for generating the figures have been deposited in the public repository: https://github.com/jz0229/PE-CMOS.

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

## Acknowledgements

We thank S. Chin and G. Brown for discussion and development of additional sparse devolution algorithms. We thank J. Liu, G.F. Lynch, and A. Khalifa for comments on the manuscript. The chip fabrication was funded by support by the Louis B. Thalheimer Fund for Translational Research, and the NIH (R21 EY028381-01). JZ, ZC and MAW acknowledge additional funding through the NIH (MH118928). ZW acknowledges the Alana Foundation. ESB acknowledges HHMI, Lisa Yang, John Doerr, NIH 1R01MH123977, NIH R01DA029639, NIH R01MH122971, NIH RF1NS113287.

## Author contributions

J.Z., J.P.N., M.A.W. conceptualize the ideas. J.Z., J.P.N. and C.L. conducted proof of concept testing. J.Z. designed the image sensor with input from E.F. and R.E.C. J.Z. and JPN implemented the acquisition hardware and firmware to steam the imaging data. J.Z., J.P.N., Z.W., and Y.Q. conducted the in vitro experiments using cultured neurons. P.F.R. conducted the patch clamp recording experiment with help from T.H. J.Z., W.G. and Z. C. processed the imaging data. All authors contributed to the manuscript. J.Z. wrote the manuscript with input from all authors. M.A.W. and E.S.B. supervised the project.

## Competing interests

The authors declare no competing interests.
