## [Peer Review File · Nature Communications]

Pixel-wise programmability enables dynamic high-SNR cameras for high-speed microscopyREVIEWER COMMENTS

Reviewer #1 (Remarks to the Author):

The authors introduce an innovative imaging sensor featuring per-pixel customizable sampling speed and phase, tailored for widefield fluorescence microscopy. This sensor is employed to capture fluorescent signals emanating from neurons that express genetically encoded voltage indicators (GEVI). Designed and fabricated using a 180 nm process supplied by XFAB, the sensor's performance is benchmarked against the renowned Hamamatsu Orca Flash 4.0 v3 sCMOS, a staple in the neuroscience community. Given the compactness of this sensor, it offers the potential for recording fluorescence from neurons in animals that are free to move.

There are few major concerns that the authors need to address:

1. The authors contend that utilizing a 2 by 2 pixel neighborhood combined with staggered sampling (or phase offset exposure) results in a superior SNR than an sCMOS camera with equivalent pixel dimensions. Central to this claim is the idea that their proposed sensor can have an exposure duration four times lengthier than that of the sCMOS, with the phase-offset sampling bolstering their SNR.

When comparing the proposed sensor's 2 by 2 pixel configuration, it seems logical to contrast it against an sCMOS camera with pixel covering an area four times larger and maintaining the same integration duration – essentially, an exposure time that's 4 times shorter than the PE-CMOS. This ensures a consistent number of photons striking the observation area, whether that's a single pixel in the sCMOS or a 2 by 2 grid in the PE-CMOS. Additionally, the sCMOS camera, due to its reduced exposure time, would accumulate similar photon count, leading to a faster sampling rate and higher SNR.

This comparison and the ensuing discussion should be integrated into Figure 1 and be reflected in the accompanying text. To support this argument with empirical evidence, the introduction of the necessary magnification between the beam splitter and the sCMOS output (a 2x optical boost) would be beneficial.

2. The claim that the PE-CMOS camera outperforms the sCMOS in capturing subthreshold events requires validation through electrophysiology. Without this, it remains ambiguous whether the observed data is a genuine neural event or merely an artifact arising from the recording process, such as phase offset, interpolation, and the like.

3. The description of the interpolation method in the text is somewhat lacking in clarity. A more rigorous theoretical analysis (or relevant citations) is needed to elucidate how interpolating multiple low-frequency samples with phase offset can produce high-frequency data. Under what circumstances is this theory valid, and when might it fall short? Adding or interpolating multiple uncorrelated samples could increase the overall noise, potentially negating the advantages of extended exposure times.

Here are few minor concerns that should be addressed:

1. The sCMOS images in Fig. 2 A and D appear noticeably noisier than the pe-CMOS image. Yet, this isn't observed in Figure 2G. The individual time plots similarly show heightened noise in the sCMOS images for A and D, but not for image G. What accounts for this discrepancy? It gives the impression that the sCMOS settings varied, which should be clearly mentioned in the manuscript.

2. The authors introduced two distinct sensors but failed to articulate the differences or the rationale behind creating both. An elaboration on this is required. Additionally, it should be clarified in the manuscript which of these sensors was utilized for the comparative analysis with the sCMOS sensor.

3. The authors should provide data detailing the conversion gain and quantum efficiency of their

sensor, and juxtapose this data with that of the sCMOS camera. This is crucial, especially considering that the text makes SNR comparisons premised on the equivalence of the two sensors. Additionally, ensure that the read-out noise can be deduced from the conversion gain.

Reviewer #2 (Remarks to the Author):

Summary

In their manuscript titled: "Pixel-wise programmability enables dynamic high-SNR cameras for high-speed microscopy", Zhang et al present a novel CMOS chip design to control the sampling frequency and phase offset of individual pixels independently. To that end, the authors designed and manufactured a new PE-CMOS design. The paper showcases two variants, Chip A (6.5 μm , 240 x 352 pixels) and Chip B (10 μm , 128 x 352 pixels).

While imaging cells in biological tissue, this approach make it possible to adapt light collection to the inhomogenous distribution of fluorophore and even out Signal to Noise Ratios (SNRs). In addition, by combining different phases and/or sampling frequency in pixels of the same Region of Interest (ROI), they demonstrate improved detection of temporal dynamics, beyond the original sampling frequency of each individual pixels. They apply this methodology to voltage imaging as well as some non-biological problems for general natural scene video acquisitions.

Generally, the application of such sensors to voltage imaging is very novel and could be impactful if technology adoption is proven to be practical. I felt this work could complement other efforts to enable large-scale voltage imaging and is a valuable addition to the existing literature. The manuscript was overall clear and well-written.

MAJOR COMMENTS

1. The design principles and circuit operation of the design were described but there are little details on how exactly to replicate manufacturing of this sensor design. In addition, it wasn't entirely clear if the design of the sensor was available. I also could not find description of how the sensor was controlled. Was an additional circuit board required? Is there custom software involved? It could help replication to provide those details.
2. Figure 1D is where the key technological innovation is introduced. I feel like this could be described in more depth in the main text, perhaps in a way that can be understood by a broader audience. The majority of the main text is application and does not fully explain the technological innovation.
3. I felt there could be more discussion on the limits of this technology. For example, the development of voltage imaging is perhaps highly geared toward in vivo monitoring of neuronal activity. Could the author comment on the impact of such inhomogeneous sampling strategy in the context of motion artifacts that are present with live samples? Does the sample movement impact more sophisticated processing routines involved in extracting the signal? More generally, what are other challenges introduced in data processing in relation to this unique data acquisition scheme?
4. Modern sCMOS and qCMOS camera have very good read noise profiles, in some cases enabling photon counting in certain application. The Orca Flash 4.0 has been replaced with more recent generations that could have better noise profile properties. I suggest perhaps to discuss and even to provide numerical comparison with those more recent sensors.

MINOR COMMENTS

1. "head-fixed animals using high-performance but bulky microscopes and were limited to a few minutes in duration^{5,6}",
This is not entirely true as there are now recent voltage imaging publication that have reported longer imaging in vivo. See Platasa et al 2023
2. There is additional prior art related to laser power excitation adapting to the sample. I suggest citing Yang et al 2017 (<https://doi.org/10.1038/s41598-017-06065-7>). Perhaps how the proposed technique differs could be discussed.
3. "However, the standard CMOS image sensors do not have flexible pixel-level exposure

programmability to allow such configuration.”

At this point, the configuration referred to was not entirely explained. I suggest revising to introduce the “configuration” earlier.

4. “In terms of temporal resolution, it is essential to obtain a minimum of three samples to accurately represent the shape of a spike over time, with one taken at the peak and two at the troughs”.

While the authors provided extensive simulation, this sentence lacked precision. Surely, three samples are not enough to identify the peak of a spike which is very brief. There are obviously trade-offs in spike detection and temporal precision here. This sentence felt like an over-simplification of the challenge at hand. I suggest being more measured in how many samples are needed.

We are grateful to the reviewers for dedicating their time to thoroughly review our manuscript. Your insightful comments have been instrumental in substantially enhancing the quality of our work.

Each reviewer's comment is highlight in gray. Our responses are in black font. The brown font are quotes from the revise manuscript that reflects the added/change text in response to reviewer's comments.

Reviewer #1 (Remarks to the Author):

The authors introduce an innovative imaging sensor featuring per-pixel customizable sampling speed and phase, tailored for widefield fluorescence microscopy. This sensor is employed to capture fluorescent signals emanating from neurons that express genetically encoded voltage indicators (GEVI). Designed and fabricated using a 180 nm process supplied by XFAB, the sensor's performance is benchmarked against the renowned Hamamatsu Orca Flash 4.0 v3 sCMOS, a staple in the neuroscience community. Given the compactness of this sensor, it offers the potential for recording fluorescence from neurons in animals that are free to move.

There are few major concerns that the authors need to address:

1. The authors contend that utilizing a 2 by 2 pixel neighborhood combined with staggered sampling (or phase offset exposure) results in a superior SNR than an sCMOS camera with equivalent pixel dimensions. Central to this claim is the idea that their proposed sensor can have an exposure duration four times lengthier than that of the sCMOS, with the phase-offset sampling bolstering their SNR.

When comparing the proposed sensor's 2 by 2 pixel configuration, it seems logical to contrast it against an sCMOS camera with pixel covering an area four times larger and maintaining the same integration duration – essentially, an exposure time that's 4 times shorter than the PE-CMOS. This ensures a consistent number of photons striking the observation area, whether that's a single pixel in the sCMOS or a 2 by 2 grid in the PE-CMOS. Additionally, the sCMOS camera, due to its reduced exposure time, would accumulate similar photon count, leading to a faster sampling rate and higher SNR.

This comparison and the ensuing discussion should be integrated into Figure 1 and be reflected in the accompanying text. To support this argument with empirical evidence, the introduction of the necessary magnification between the beam splitter and the

We understand the purpose of this comparison is to compare the pixel performance by equating the number of photons in 1 sCMOS pixels with that of 4 PE-CMOS pixels. This could be accomplished by making the optical magnification on the PE-CMOS larger, such that 4 pixels of PE-CMOS can be compared with 1 single sCMOS pixel (Fig. R1). In other words, 1 sCMOS pixel covers 4x larger area than 1 PE-CMOS pixel.

The difficulty with this comparison is that in our GEVI imaging videos, a single pixel's SNR is relatively low. In Fig. R2, we plotted the time series of a single sCMOS pixel located on the membrane of the cell in Fig. 2A. With such low SNR, it is not possible to carry out pixel-to-pixel comparisons across two sensors or compare 1 sCMOS pixel with 4 PE-CMOS pixels.

Fig. R1. Equating the number of photons in 1 sCMOS pixel with that from 4 PE-CMOS pixels

Instead, we could average many sCMOS pixels' outputs within the ROI to enhance SNR and compare them against 4x more PE-CMOS pixels. In this case, we are essentially comparing the average signal generated from photons over a ROI. We would argue that this comparison would be the same as the ones we have conducted in the manuscript (Fig. R3). In Fig. 2A-F of the manuscript, for the sCMOS outputs, we evenly split the ROI into 4 sections and compared 1/4 ROI's average time series with the PE-CMOS at different phases without interpolation (each also consists of pixels from 1/4 ROI). We then compared the average signal from all pixels in the ROI across two sensors.

Fig. R2. For the neuron in Fig. 2A, a single sCMOS pixel time-series has much lower SNR than the ROI average.

Fig. R3. To compare the signal generated over an ROI, the optical magnification should not affect the result of the comparison.

We added the text below to the revised manuscript to explain the rationale of the comparison:

We compared the PE-CMOS with the sCMOS outputs over the same ROI (Fig. 2B, C, E, F). To ensure a fair comparison of signals from equivalent areas, we evenly divided the sCMOS pixel in the ROI into four sections (Fig. 2B, E). We then compared the average time series from each section against the PE-CMOS outputs at different phases (Fig. 2C, F), with each phase covering 1/4 of the pixels of ROI. Following this, we compared the interpolated PE-CMOS signal over the entire ROI with sCMOS output averaged over the equivalent ROI.

2. The claim that the PE-CMOS camera outperforms the sCMOS in capturing subthreshold events requires validation through electrophysiology. Without this, it remains ambiguous whether the observed data is a genuine neural event or merely an artifact arising from the recording process, such as phase offset, interpolation, and the like.

Yes, we agree with the reviewer's comment. We can demonstrate this experimentally by recording the intracellular potential of a cultured neuron expressing the GEVI indicator (ASAP3), while we image the neuron using both the PE-CMOS and sCMOS sensor. We added a new figure (new Fig. 3) to the manuscript to describe this result.

Since the microscopy setup we used previously was repurposed for another project. We had to set up a new patch clamp rig for this experiment. All imaging conditions are the same except for the microscope objective lens. We used a 40x/0.6NA (Leica) objective lens for simultaneous imaging and patch clamp experiment, whereas all the previous imaging experiments used a 20x/1.0 NA (Olympus) lens. This difference is explained in detail in the Methods section.

We also added the following text to the results section of the revised manuscript to describe the result of this new experiment.

To make a direct comparison for capturing somatic voltage, we used a patch clamp to measure the intracellular potential while performing simultaneous GEVI imaging with both sCMOS and PE-CMOS cameras (**Fig. 3**). To excite cells during imaging, we injected 200 ms duration current pulses of various amplitudes (+600 pA to 0 pA with gradually decreasing 40 pA steps). To quantify the GEVI signal measured by PE-CMOS and sCMOS, we can measure the GEVI response associated with each current injection pulse. The GEVI pulse amplitude is defined as the difference between the average GEVI intensity during each current pulse and the average GEVI intensity 100 ms before and after the pulse (**Fig. 3A**). GEVI pulse amplitudes are converted into SNR by dividing the noise standard deviation. The noise standard deviation is measured from the GEVI intensity in the absence of current injection pulses. GEVI pulse amplitudes in the PE-CMOS are consistently higher than in sCMOS (**Fig. 3A**, bar plot). This can be explained by reference to the frequency response of the PE-CMOS (**Supplementary Fig. 3E**). Due to 4x prolonged pixel exposure, the PE-CMOS applies a high amount of gain to a lower frequency signal than sCMOS, which has a uniform gain profile across the frequency range.

However, one could ask, would sampling at 200 Hz with sCMOS achieve the same results? We can mimic an $F_s = 200$ Hz sCMOS signal by convolving an $F_s = 800$ Hz signal with a 4 ms box function followed by downsampling by a factor of 4. This would filter out the noise at high frequencies, which increases the SNR of low-frequency GEVI pulse (**Fig. 3C,D**). However, the resulting signal has a temporal resolution of only 200 Hz, which is insufficient to capture spiking activities, especially at resolving spikes with low inter-spike intervals. We can identify spike positions (**Fig. 3C**, red arrows) with electrophysiology recordings and examine the corresponding GEVI signal amplitude from the PE-CMOS and sCMOS cameras. At $F_s = 800$ Hz, the sCMOS signal exhibits low SNR, making the spikes less distinguishable from noise than those captured by the PE-CMOS. By filtering high-frequency noise, the sCMOS signal at 200 Hz improves the SNR of some of the spikes. However, this reduced sampling rate leads to aliasing, negatively impacting the amplitudes of other spikes, particularly those with short inter-spike intervals (**Fig. 3C, D**). The sCMOS output at $F_s = 200$ Hz has a Nyquist resolution (defined as $2/F_s$) of 10 ms. In this case, spikes with inter-spike intervals of 17.4 and 19.5 ms are aliased, causing the spike amplitude to decrease drastically to the point that it can no longer be resolved (**Fig. 3D**, red arrows). On the other hand, PE-CMOS minimizes the aliasing effect with a 2.5 ms Nyquist resolution, 4x time better than sCMOS at $F_s = 200$ Hz, preserving the spike amplitude. The ability of PE-CMOS to avoid aliasing is also replicated in **Supplementary Fig. 3**, with a controlled experiment using an LED to produce optical spike signals spatially uniform across the sensor.

It is also important to note that among the spikes invoked by the current injection pulse, the first spike (shown in **Fig. 3D** and marked with blue arrows) poses a significant sampling challenge for the image sensors. This is due to its high-frequency component, which is attributed to the sharp rising edges. Capturing these spikes with sCMOS sensor sampling at 800 Hz leads to low SNR (**Fig. 3D**). Sampling it at 200 Hz leads to aliasing, which decreases the spike's amplitude (**Fig. 3D**). In contrast, the PE-CMOS preserves these high-frequency components more effectively than sCMOS (**Fig. 3D**). This illustrates the PE-CMOS's advantage at capturing high-frequency signals in noisy conditions.

We also agree with the reviewer's comment that we need to be mindful of the potential artifacts that might arise. We address this with the following text in the manuscript.

In this experiment, one potential ambiguity might be the GEVI signals at the end of some current pulses (**Fig. 3C**, black arrow). While intracellular potential shows a flat response, the GEVI signals in both PE-CMOS and sCMOS exhibit significant amplitude variations, which could be mistaken for spiking events. Given that this phenomenon is observed in the outputs from both PE-CMOS and sCMOS sensors (at 800 and 200 Hz), we believe this is not an artifact specific to the PE-CMOS sensor. Instead, it likely results from the response of the GEVI indicators. To ensure the PE-CMOS's interpolation process does not introduce systematic artifacts, we examine the interpolation process in detail (**Methods**), even when interpolating an ROI with pixels of

uncorrelated activity. We showed that interpolating uncorrelated pixels yields an approximation of their average. The selection of regression parameters in the interpolation process minimizes the overfitting of noise.

We will further address the reviewer's concerns regarding phase offset and interpolation effects in our subsequent response.

3. The description of the interpolation method in the text is somewhat lacking in clarity. A more rigorous theoretical analysis (or relevant citations) is needed to elucidate how interpolating multiple low-frequency samples with phase offset can produce high-frequency data. Under what circumstances is this theory valid, and when might it fall short? Adding or interpolating multiple uncorrelated samples could increase the overall noise, potentially negating the advantages of extended exposure times.

We first include a theoretical derivation of the interpolation process for an ROI with a correlated signal, showing how interpolating phase-shifted samples at a low sampling rate can avoid aliasing and improve the temporal resolution of the resulting signal. We then derive the interpolation steps for the signal and use an example to show how our interpolation steps avoid overfitting. The following are included in the supplemental methods:

Theoretical analysis on how multiple low-frequency samples with phase offset can avoid aliasing: we include the following text in the method section of the revised manuscript, together with a new Fig. S8.

In an ROI with highly correlated pixel outputs, we demonstrate that combining the outputs from low-frequency pixels, each sampled at distinct phase-shifted exposures, can eliminate aliasing and improve the temporal resolution of the ROI average time series.

We assume a light impulse, $\delta(t)$, is simultaneously sampled by 4 pixels (**Supplemental Fig. 8A**), y_1, \dots, y_4 with an exposure function $e(t)$:

$$y_k(t) = e(t) * \delta(t), \quad e(t) = \begin{cases} 1 & 0 \leq t \leq T_E \\ 0 & \text{otherwise} \end{cases}, \quad k \in \{1,4\},$$

$y_k(t)$ is then sampled by an ADC at a period of T_E . T_E is also the exposure duration. We then write the discretized version of $y_k(t)$:

$$y_k[n] = y_k(t) \cdot \sum_{n=-\infty}^{\infty} \delta(t - nT_E)$$

with frequency spectrum:

$$Y_k(f) = \frac{1}{T_E} \sum_{n=-\infty}^{\infty} E(f - \frac{n}{T_E})$$

Where $E(f - \frac{n}{T_E})$ are the replicas of the exposure function's spectrum resulting from sampling. We can plot $Y_k(f)$ to see that the majority of the frequency spectrum is aliased, resulting in signal distortion (**Supplemental Fig. 8B**).

By relying on the phase difference between neighboring pixels, we can eliminate the aliasing effect without increasing each pixel's sampling speed. To achieve this, relative to the phase of pixel 1, we can introduce phase shifts of $\frac{T_E}{4}, \frac{T_E}{2}, \frac{3T_E}{4}$ to pixels 2, 3, and 4, respectively (**Supplemental Fig. 8C**). These pixels' spectrum, $Y_k(f)$, becomes:

$$Y_k(f) = \frac{1}{T_E} \sum_{m=-\infty}^{\infty} E\left(f - \frac{n}{T_E}\right) e^{-j2\pi \frac{n(k-1)}{4}}$$

where $k \in \{1,4\}$. If we average these four pixels, the resulting spectrum becomes:

$$Y_{avg}(f) = \frac{1}{4} \sum_{k=1}^4 Y_k(f) = \frac{1}{4T_E} \sum_{m=-\infty}^{\infty} E\left(f - \frac{n}{T_E}\right) (1 + e^{-j2\pi \frac{n}{4}} + e^{-j2\pi \frac{n}{2}} + e^{-j2\pi \frac{3n}{4}})$$

since $Y_{avg}(f) = 0$ when m is not a multiple of 4, the above equation can be rewritten as:

$$Y_{avg}(f) = \frac{1}{4T_E} \sum_{n=-\infty}^{\infty} E\left(f - \frac{4n}{T_E}\right)$$

which spaces the frequency replicas of $E(f)$ away from each other and avoids aliasing (**Supplemental Fig. 8D**). The average spectrum of phase-shifted pixels increases the Nyquist bandwidth of individual pixels by four times, without an increase in per-pixel sampling rate. From a time-domain viewpoint, the combined phase-shifted pixel outputs are equivalent to the sampling of the signal, $v(t)$, at a higher rate of $4/T_E$.

Interpolating uncorrelated signals: Sections of the revised text below were in the original manuscript's supplementary methods under "ROI signal interpolation". In the revised text, we included more details and examples (a new Supplemental Fig. 9) to explain the rationale behind the parameter choice to avoid overfitting the noise and demonstrate that interpolating uncorrelated pixels will approximate the ROI average.

We employ linear interpolation to compute the ROI time series from the pixel outputs. For an ROI containing N pixels, let \mathbf{v}_n , $n = \{1 \dots N\}$, of discrete length L represent the fluorescence signal at n th pixel. The sampled value at this pixel, \mathbf{y}_n , of discrete length, M is:

$$\mathbf{y}_n = \mathbf{E}_n \mathbf{v}_n \quad (5)$$

Where $\mathbf{E}_n \in \mathbb{R}^{M \times L}$ denotes the sampling matrix representing pixel exposure and sampling operation that converts the fluorescent signal $\mathbf{v}_n \in \mathbb{R}^L$ into the pixel outputs, $\mathbf{y}_n \in \mathbb{R}^M$.

Now, we can define an arbitrary signal, \mathbf{v}_{ROI} , and rewrite each \mathbf{y}_n as:

$$\mathbf{y}_n = \mathbf{E}_n \mathbf{v}_{ROI} + \mathbf{E}_n (\mathbf{v}_n - \mathbf{v}_{ROI})$$

Note that we do not make the assumption that \mathbf{v}_{ROI} is the average of the pixels.

if we write the difference term $\boldsymbol{\varphi}_n = \mathbf{E}_n (\mathbf{v}_n - \mathbf{v}_{ROI})$, we can then concatenate the output of all of the N pixels into one vector:

$$\mathbf{y} = \mathbf{E} \mathbf{v}_{ROI} + \boldsymbol{\varphi} \quad (6)$$

$$\text{where } \mathbf{y} = \begin{bmatrix} \mathbf{y}_1 \\ \vdots \\ \mathbf{y}_N \end{bmatrix}, \mathbf{E} = \begin{bmatrix} \mathbf{E}_1 \\ \vdots \\ \mathbf{E}_N \end{bmatrix} \text{ and } \boldsymbol{\varphi} = \begin{bmatrix} \boldsymbol{\varphi}_1 \\ \vdots \\ \boldsymbol{\varphi}_N \end{bmatrix}$$

Given **Eq. 6**, we can find \mathbf{v}_{ROI} using a ridge regression by enforcing a L_2 norm penalty:

$$\mathbf{v}_{ROI} = \min_{\mathbf{v}_{ROI}} \frac{1}{2} \|\mathbf{y} - \mathbf{E} \mathbf{v}_{ROI}\|_2^2 + \lambda_{ridge} \|\mathbf{v}_{ROI}\|_2^2$$

and further derive its closed-form solution:

$$\hat{\mathbf{v}}_{ROI} = (\mathbf{E}^T \mathbf{E} + \lambda_{ridge} \mathbf{I})^{-1} \mathbf{E}^T \mathbf{y} \quad (7)$$

where $\hat{\mathbf{v}}_{ROI}$ is the least-squared estimate of \mathbf{v}_{ROI} , and the hyperparameter, λ_{ridge} , controls the trade-off between the L₂-norm of \mathbf{v}_{ROI} and reconstruction error, denoted by $\|\mathbf{y} - \mathbf{E}\mathbf{v}_{ROI}\|_2 = \|\boldsymbol{\varphi}\|_2$

Solving this regression should push \mathbf{v}_{ROI} to be close to the average of the pixels ($\overline{\mathbf{v}}_{ROI} = \frac{1}{N} \sum_{n=1}^N \mathbf{v}_n$), which by definition, minimizes the L2 norm of $\|\mathbf{y} - \mathbf{E}\mathbf{v}_{ROI}\|_2$. However, in the presence of noise, setting λ_{ridge} could have a risk of overfitting. To avoid this, in the interpolation method used in our manuscript, we set λ_{ridge} to be large, which conservatively underfit our approximation for \mathbf{v}_{ROI} .

With a large λ_{ridge} , the term $(\mathbf{E}^T \mathbf{E} + \lambda_{ridge} \mathbf{I})^{-1}$ will approximate a scaled identity matrix, and **Eq. 7** becomes

$$\hat{\mathbf{v}}_{ROI} \approx \mathbf{E}^T \mathbf{y} \quad (8)$$

which is equal to up-sampling and interpolation of the pixel outputs \mathbf{y} .

We can reinforce our analysis with a simple example in **Supplemental Fig. 9**. Here, the time series of 4 pixels, $\mathbf{v}_1 \dots \mathbf{v}_4$, contains uncorrelated signal, with \mathbf{v}_{ROI} representing the average of these signals. We mimic the exposure and phase-shifted sampling of these pixels to get $\mathbf{y}_1 \dots \mathbf{y}_4$. We do this by convolving phase-shifted version of $\mathbf{v}_1 \dots \mathbf{v}_4$ with a box function of length 4, followed by 4x down-sampling. We can see that down-sampling aliases the high-frequency part of the signal of \mathbf{y}_2 , especially at the peak of the spike (**Supplemental Fig. 9** black arrow).

We can apply our interpolation outlined in **Eq. 8**, which results in $\hat{\mathbf{v}}_{ROI}$. We can see that $\hat{\mathbf{v}}_{ROI}$ is proportional to \mathbf{v}_{ROI} , but underfits the spike at the location pointed by the black arrow in **Supplemental Fig. 9**. This is because we only sampled \mathbf{y}_2 at a single phase since \mathbf{y}_2 is uncorrelated with other pixels.

This example shows that even with uncorrelated pixels, interpolating them yields an approximation of the ROI average. As shown, the selection of large λ_{ridge} in the ridge regression minimizes the overfitting of noise during interpolation.

Here are few minor concerns that should be addressed:

1. The sCMOS images in Fig. 2 A and D appear noticeably noisier than the pe-CMOS image. Yet, this isn't observed in Figure 2G. The individual time plots similarly show heightened noise in the sCMOS images for A and D, but not for image G. What accounts for this discrepancy? It gives the impression that the sCMOS settings varied, which should be clearly mentioned in the manuscript.

Thank you for bringing up this concern. The discrepancy is due to grayscale normalization of the plot. The grayscale in these images are normalized to enhance visibility. Since these images are maximum projection images (averaged over all frames), they should have minimal readout and shot noise contribution. The static noise pattern shown is mostly the pixel array's fixed-pattern noise. In Fig. 2AD, the maximum pixel values are much smaller than Fig. 2G. This is due to the difference in GEVI expression level on the cells. As a result of grayscale normalization, the fixed pattern noise is more prominently visible. You can also see the PE-CMOS's fixed pattern noise in Fig. 2G, but they are not as visible compared to Fig. 2AD due to the normalization.

2. The authors introduced two distinct sensors but failed to articulate the differences or the rationale behind creating both. An elaboration on this is required. Additionally, it should be clarified in the manuscript which of these sensors was utilized for the comparative analysis with the sCMOS sensor.

Yes, we modified the manuscript text to be clearer on the design of these two prototypes. The different pixel pitches in the Chip A and B prototypes were to test different pixel designs and configurations because it was extremely difficult to model the full response properties of a given design until it was fabricated. Referring to Fig. S1 of the manuscript, the pixel circuits of Chip A and B are identical. The chip level circuits are slightly different, in pixel A, $K = 8$ rows share a single group of row select signals (RST, SEL, and TX), whereas, in Chip B, $K = 4$ rows share a single group of row select signals. The design of PE-CMOS is such that when K rows are selected, for 1 column, only 1 pixel per K rows can be sampled. The pixels not sampled continue their exposure. Therefore, different K sizes make a trade-off of the spatiotemporal configuration of the pixels, and we are currently working on determining the best trade-offs. Both chips functioned as expected. We used Chip B for the comparison with the sCMOS sensor in our manuscript, because it has a better fill factor (and QE) and should be better for low-light neural imaging applications. The following is added to ensure clarity:

In the subsequent experiments, we used the PE-CMOS Chip B prototype with a $10\ \mu\text{m}$ pixel pitch. It is chosen over Chip A because it has a better fill factor, which should translate to better performance for low-light imaging applications.

3. The authors should provide data detailing the conversion gain and quantum efficiency of their sensor, and juxtapose this data with that of the sCMOS camera. This is crucial, especially considering that the text makes SNR comparisons premised on the equivalence of the two sensors. Additionally, ensure that the readout noise can be deduced from the conversion gain.

Thank you for this suggestion. We have included conversion gain and noise measurement in Fig. S1. We also modified the manuscript so that they are in more prominent locations. The conversion gain and readout noise are extracted by measuring the photon transfer curve. The measured conversion gain ($110\ \mu\text{V}/e^-$) and readout noise ($2.67\ e^-$, measured at room temperature without active cooling) are listed in Fig. S1. The quantum efficiency of the photodiode is 90% at 560nm, extracted from data provided by XFAB, our chip foundry. Since our pixel has a 75% photodiode fill factor, this translates to $\sim 68\%$ pixel QE without on-chip microlens. For comparison, the Hamamatsu Orca-Flash 4.0 V3 sCMOS with $6.5\ \mu\text{m}$ pitch has 82% fill factors, with $1.6\ e^-$ read noise measured at negative 16 deg C.

Although the PE-CMOS has slightly inferior QE and noise level, these parameters can be further improved without modification to the on-chip design. We could enhance pixel QE to a level comparable to the sCMOS pixels by using on-chip microlens fabricated using commercial fabrication processes. In addition, our read noise is measured at room temperature, which could be further decreased by cooling the sensor, similar to the sCMOS camera design. The follow text is added to compare with sCMOS.

It is also worth noting that due to longer pixel exposure, time-staggered sampling using PE-CMOS outperformed sCMOS, despite its slightly worse read noise performance and lower QE (readnoise: $2.67\ e^-$ at room temperature, with QE of 68% without microlens), compared to that of the sCMOS (datasheet readnoise: $1.6\ e^-$ at -16 deg. C, with QE of 82%).

Reviewer #2 (Remarks to the Author):

Summary

In their manuscript titled: “Pixel-wise programmability enables dynamic high-SNR cameras for high-speed microscopy”, Zhang et al present a novel CMOS chip design to control the sampling frequency and phase offset of individual pixels independently. To that end, the authors designed and manufactured a new PE-CMOS design. The paper showcases two variants, Chip A (6.5 um, 240 x 352 pixels) and Chip B (10 um, 128 x 352 pixels).

While imaging cells in biological tissue, this approach make it possible to adapt light collection to the inhomogenous distribution of fluorophore and even out Signal to Noise Ratios (SNRs). In addition, by combining different phases and/or sampling frequency in pixels of the same Region of Interest (ROI), they demonstrate improved detection of temporal dynamics, beyond the original sampling frequency of each individual pixels. They apply this methodology to voltage imaging as well as some non-biological problems for general natural scene video acquisitions.

Generally, the application of such sensors to voltage imaging is very novel and could be impactful if technology adoption is proven to be practical. I felt this work could complement other efforts to enable large-scale voltage imaging and is a valuable addition to the existing literature. The manuscript was overall clear and well-written.

MAJOR COMMENTS

1. The design principles and circuit operation of the design were described but there are little details on how exactly to replicate manufacturing of this sensor design. In addition, it wasn't entirely clear if the design of the sensor was available. I also could not find description of how the sensor was controlled. Was an additional circuit board required? Is there custom software involved? It could help replication to provide those details.

Thank you for suggesting this to improve the manuscript's clarity. We had the circuit's operation details in the Methods section of the original manuscript. In the revised manuscript, we have re-arranged the sections so that they are in the main text. Please refer to the “PE-CMOS circuit architecture” under the results section.

With respect the chip itself, we are happy to provide all the information required to replicate the design. However, the design documents use the proprietary circuit models from the commercial foundry protected under the NDA. We can share the design with the university after they have secured the same NDA with XFAB. We currently have chip samples we could give to the interest parties upon request.

To interface with the chip, we used open-source hardware, firmware, and software API from the Open-Ephys ONIX¹⁶ project and Bonsai¹⁷.

We added explanations on the sensor availability and data sharing at the end of the manuscript under “data and design availability.”

2. Figure 1D is where the key technological innovation is introduced. I feel like this could be described in more depth in the main text, perhaps in a way that can be understood by a broader audience. The majority of the main text is application and does not fully explain the technological innovation.

Thank you for pointing this out. We have described Figure 1D more in detail in the new section (“PE-CMOS circuit architecture”) of the revised manuscript.

3. I felt there could be more discussion on the limits of this technology. For example, the development of voltage imaging is perhaps highly geared toward *in vivo* monitoring of neuronal activity. Could the author comment on the impact of such inhomogeneous sampling strategy in the context of motion artifacts that are present with live samples? Does the sample movement impact more sophisticated processing routines involved in extracting the signal? More generally, what are other challenges introduced in data processing in relation to this unique data acquisition scheme?

The reviewer raised an important issue we might encounter during *in vivo* applications. We plan to follow up on the *in vivo* experiment in our future work. Below are our thoughts on PE-CMOS's expected advantages and limitations in correcting for cell motion.

It has been shown that during *in vivo* behavioral experiments, the cells can experience small movements in the FOV. To correct this, we can track the cell position at each frame and re-align the cell in the ROI through x-y position transformation. Accurate motion tracking depends on (1) temporal resolution: how accurately we can sample the cell motion, and (2) motion blurring: how accurately we can isolate the exact position of the cell. Although we did not test the PE-CMOS under an *in vivo* setting, **Supplemental Fig. 6C, D** provides an intuitive example of tracking object movement in the PE-CMOS sensor.

In this example, we sampled a bouncing ball's trajectory at a 25 Hz pixel sampling rate (40 ms exposure) at four phase offsets (**Supplemental Fig. 6C, D**). This arrangement effectively increases the temporal resolution of tracking this motion to 100 Hz. As a result, we could capture the motion much more accurately (**Supplemental Fig. 6E**) compared to a conventional sCMOS camera without phase-offsets at a 25 Hz sampling rate.

However, the motion blurring is governed by the pixel exposure time. As a result, we can see more blurring at high movement speeds (**Supplemental Fig. 6D**). We could compensate for the blurring by enlarging our ROI boundary to encompass the blurred region. But, for cells located nearby, the blurring could cause ambiguity in ROI selection, as the boundary between these two cells might become obscured.

Blurring will only become a problem for high-speed cell movement. Although we do not know the exact amount of cell movement as we have not tested the PE-CMOS sensor *in vivo*, we have not experienced blurring-induced inaccuracy in cell tracking in our calcium imaging experiments conducted on freely moving mice using the UCLA miniscope v3, which sample at 10 – 30 Hz (100 – 30 ms exposure time). Therefore, we do not expect this will be a problem for GEVI imaging, where the PE-CMOS pixel sampling rate is >200 Hz with exposure <4 ms. However, this claim needs careful validation from *in vivo* experiments.

We discussed motion blur and other potential limitations of the approach in the discussion section under "Potential limitations".

Potential limitations. We have demonstrated the performance of the PE-CMOS through *in vitro* experiments. We foresee several possible limitations that would benefit from *in vivo* validations. First, during *in vivo* imaging in freely behavioral mice, the cells can experience small movements in the FOV. To correct this movement, we can track each cell's position at each frame and re-align them through transformation. Accurate motion tracking depends on (1) temporal resolution: how accurately we can sample the cell motion, and (2) motion blurring: how accurately we can isolate the exact position of the cell. The PE-CMOS can achieve higher temporal resolution, but the blurring is determined by its pixel exposure duration. Blurring will only become a problem for high-speed cell movement. Although we do not know the exact amount of cell movement, previous calcium imaging in behaving mice offers estimates. The microscopes sample the FOV at 10 – 30 Hz (with 100 ms to 30 ms of blurring) and have not reported blurring-induced inaccuracy in motion correction. Therefore, we

do not expect this will be a problem for GEVI imaging, where the PE-CMOS pixel sampling rate is >200 Hz with exposure <4 ms. However, this claim needs careful validation from *in vivo* experiments.

Second, phase-shifted pixels enhance the bandwidth of the sampled ROI time series, but it does not eliminate the narrowband attenuations (e.g. seen at 250 and 500 Hz in **Supplemental Fig. 3E**) induced by long pixel exposures. While these attenuations may not affect the detection of spikes with a broad frequency spectrum, they may pose a problem when the ROI contains specific signals of interest (such as brain oscillations) that fall into these frequency ranges. To resolve this, the PE-CMOS can employ pixels with varied exposure durations within the ROI (**Supplemental Fig. 4**). These pixels sample the ROI at different speeds with varying exposure. While certain frequencies may be attenuated by pixels with a specific exposure duration, they are preserved by other pixels with different exposure lengths. Consequently, the frequency spectrum of the ROI displays no narrowband attenuation.

4. Modern sCMOS and qCMOS camera have very good read noise profiles, in some cases enabling photon counting in certain application. The Orca Flash 4.0 has been replaced with more recent generations that could have better noise profile properties. I suggest perhaps to discuss and even to provide numerical comparison with those more recent sensors.

We agree with the reviewer that modern sCMOS has achieved an impressive read noise profile and sensitivity. This is mainly due to advancements in CMOS fabrication processes (back-illuminated photodiode, low-noise transistors, and novel design of the photodiode doping profiles). However, the pixel circuit design has mainly remained the same. They all utilized either a global shutter or a rolling shutter for readout, which does not allow pixel-wise exposure configurations.

Compared to them, our contribution is to demonstrate a new pixel circuit design that gives rise to pixel-wise programmability. Through the phase-shifted exposures, we also demonstrated a new sampling method suitable for physiological signals of an ROI. The same circuit architecture could be used on the sCMOS or qCMOS's pixel fabrication process for even better performance.

To offer a numerical comparison, we provide the measurements of the PE-CMOS pixels in Supplemental Fig. 1: For the PE-CMOS with a $10\ \mu\text{m}$ pitch, the measured conversion gain is $110\ \text{uV/e}^-$ and readout noise is $2.67\ \text{e}^-$, measured at room temperature without active cooling. The photodiode's quantum efficiency (QE) is 90% at 560nm, extracted from data provided by XFAB. Since our pixel has a 75% photodiode fill factor, this translates to $\sim 68\%$ pixel QE without on-chip microlens. For comparison, the Hamamatsu Orca-Flash 4.0 V3 sCMOS with $6.5\ \mu\text{m}$ pitch has 82% fill factors, with $1.6\ \text{e}^-$ read noise measured at negative 16°C .

Although the PE-CMOS has slightly inferior QE and noise level, these parameters can be further improved without modification to the on-chip design. We could enhance pixel QE to a level comparable to the sCMOS pixels by using on-chip microlens fabricated using commercial fabrication processes. In addition, our read noise is measured at room temperature, which could be further decreased by cooling the sensor, similar to the sCMOS camera design.

It is also worth noting that PE-CMOS still outperformed sCMOS, despite its slightly worse read noise performance and lower QE. This is attributed to the PE-CMOS having 4x more exposure, accumulating 4x more photons. We modeled the expected spike SNR for PE-CMOS and sCMOS with at different noise levels (**Supplemental Fig. 5**). This theoretical estimation agrees with our empirically measured spike amplitude gain, where the PE-CMOS achieves a ~ 2 -3 fold gain in SNR compared to the sCMOS sensor.

We summarized this in the discussion section of the revised manuscript, under "future improvements".

Future improvements: The PE-CMOS sensor can be further improved for better sensitivity and noise performance. Without refabricating the chip, we could implement a layer of microlens array on top of the pixel array to enhance the pixel quantum efficiency, a widely used process for image sensors. On the system side, we could implement active cooling to reduce the chip temperature to achieve lower read noise. The demonstrated PE-CMOS circuits architecture is a generalized circuit design, which could be implemented using the advanced CMOS image sensor fabrication processes (such as back-illuminated photodiode processes) to further take advantage of enhanced photodiode sensitivity and low noise transistors to improve the overall optical performance.

MINOR COMMENTS

1. "head-fixed animals using high-performance but bulky microscopes and were limited to a few minutes in duration^{5,6},".

This is not entirely true as there are now recent voltage imaging publication that have reported longer imaging in vivo. See Platisa et al 2023

Since we submitted our manuscript, there have been many optical and algorithm innovations on improving imaging SNR at high speed, in addition to the citation mentioned by the reviewer. We have added a paragraph in the introduction section to describe them. See below:

While many efforts have focused on improving the optical system and novel de-noising algorithmic solutions, the key trade-offs of speed and SNR are also fundamentally linked to the camera image sensors. For example, a new optical method maximizes SNR of the ROI's intensity trace, by focusing photons from the entire ROI onto a single pixel⁷ Additionally, an innovative electro-optical modulation method allows for the direct measurement of fluorescence lifetime signals⁸, which offers enhanced signal fidelity compared to traditional fluorescence intensity measurements. On the algorithm side, advanced denoising techniques^{9,10} based on deep learning can boost SNR under low fluorescence excitation and prolong experimental time from minutes to an hour. In this work, we examine and address the limitations of SNR at high speed from a pixel and image sensor perspective. Advancements in sensor technology will complement ongoing innovations in other technical areas. Together, they will enhance our fluorescence imaging technology to enable the tracking of neural activity at millisecond resolution across a large number of neurons, over long experimental durations.

2. There is additional prior art related to laser power excitation adapting to the sample. I suggest citing Yang et al 2017 (<https://doi.org/10.1038/s41598-017-06065-7>). Perhaps how the proposed technique differs could be discussed.

Thank you for this suggestion. The technique mentioned by the reviewer has a similar general idea compared to our proposed multiple temporal resolution sampling of the ROI (**Fig. 4**). Like the adaptive power in a closed-loop, we could also put the PE-CMOS pixels in a closed-loop to achieve similar effects at the sensor side to expand the dynamic range. We actually accomplished this using a previously designed pixel-wise image sensor design for a high-dynamic range video applications (see the reference below). Adjustment of pixel exposure is perhaps more applicable to wide-field imaging, whereas the pixel-wise excitation laser power adaptation is more suitable for two-photon imaging applications.

Zhang, J., Newman, J. P., Wang, X., Thakur, C. S., Rattray, J., Etienne-Cummings, R., & Wilson, M. A. (2020). A closed-loop, all-electronic pixel-wise adaptive imaging system for high dynamic range videography. *IEEE Transactions on Circuits and Systems I: Regular Papers*, 67(6), 1803-1814.

We added text in the results section to highlight this point:

Sampling with spatially varying pixel exposures also enhances the dynamic range for FOV with large brightness variation, often caused by uneven fluorescence indicator expression. To sample these scenes, uniform pixel exposure can cause signal saturation in bright areas and low SNR in dim ones. Unlike in two-photon microscopy, adjusting excitation power per pixel¹⁸ is challenging in wide-field excitation without complex optical setups. Spatially varying pixel exposures capture a 2 x 2 pixel region with different exposures (**Fig. 4B**), simultaneously capturing bright areas with short exposures to prevent oversaturation and enhancing dim area's SNR with longer exposures. The pixel exposure can also be adaptively adjusted in a closed-loop system to optimize the dynamic range further, similar to our previously demonstrated method¹⁹.

3. "However, the standard CMOS image sensors do not have flexible pixel-level exposure programmability to allow such configuration."

At this point, the configuration referred to was not entirely explained. I suggest revising to introduce the "configuration" earlier.

Thank you for the suggestion to improve the manuscript's clarity. We have re-wrote the first 2 paragraphs of the result section:

A fundamental trade-off exists between a pixel's sampling speed and SNR. Fast sampling speeds lead to high readout noise, shortened exposure time, and fewer collected photons, inevitably lowering the SNR. The CMOS image sensor, which uniformly exposes and samples an array of pixels, is subject to the same SNR and speed limitation. High frame rates result in low SNR, and low frame rates with long pixel exposure lead to signal aliasing (**Fig. 1A, B**).

Programmable pixel-wise exposure CMOS image sensor. Despite this trade-off, physiological signals, such as the membrane voltage of a cell soma, are often spatially correlated, and the same signal can be redundantly captured by multiple pixels within a region of interest (ROI). We demonstrate a CMOS image sensor with pixel-wise programmable exposures ("PE-CMOS") to take advantage of the highly correlated nature of microscopy scenes. The PE-CMOS permits flexible exposure at each pixel. This feature allows versatile pixel configurations to increase temporal resolution at sampling physiological signals without sacrificing SNR. In one configuration, the PE-CMOS staggered pixels' exposure in time to acquire fast-spiking events at multiple phases (**Fig. 1C**), resolving temporal details finer than the exposure times. Importantly, this increase in temporal resolution is achieved without raising the pixel sampling rate or reducing exposure time, therefore avoiding sacrificing the SNR. In another configuration, the PE-CMOS samples the ROI with pixels at different speeds, capturing high-frequency events (spiking activity) and weak signals (subthreshold potentials) that are difficult to acquire simultaneously at a fixed frame rate (**Fig. 1C**). The flexible pixel-wise exposure configuration is not achievable in conventional CMOS architecture, as all the pixels must have the same exposure, limited by the frame rate, and are sampled concurrently (global shutter) or sequentially in lines (rolling shutter).

4. "In terms of temporal resolution, it is essential to obtain a minimum of three samples to accurately represent the shape of a spike over time, with one taken at the peak and two at the troughs".

While the authors provided extensive simulation, this sentence lacked precision. Surely, three samples are not enough to identify the peak of a spike which is very brief. There are obviously trade-offs in spike detection and temporal precision here. This sentence felt like an over-simplification of the challenge at hand. I suggest being more measured in how many samples are needed.

Yes, we agree with that this was an oversimplified explanation. We think these two concepts (temporal resolution and SNR) should be defined separately to be consistent with the rest of the paper. For sCMOS, the 1.25 ms exposure duration determines the temporal resolution and SNR. In the PE-CMOS, the temporal resolution is determined by the temporally shifted pixels (1.25 ms). But the SNR is determined by the exposure duration (5 ms). For spike detection, both factors are important. The temporal resolution determines the spike

peak position, while SNR improves the spike amplitude captured at the sensor. We have modified the text to reflect this discussion.

We set the sCMOS camera to sample signal with 1.25 ms exposure duration, which also determines its temporal resolution and sampling rate (F_s) of 800 Hz. In the PE-CMOS camera, we used a longer exposure time of 5 ms to integrate over the full spike width (half-width of the action potential generated by ASAP3 spikes is ~ 6 ms). We shift pixel exposures at phase offsets in multiples of 1.25 ms (**Fig. 1C**). This sets the PE-CMOS temporal resolution to be the same as the sCMOS at 1.25 ms. Although each PE-CMOS pixel has a low sampling speed of 200 Hz, the PE-CMOS can acquire the ROI at an equivalent of 800 Hz with pixels at different phases.

REVIEWERS' COMMENTS

Reviewer #1 (Remarks to the Author):

The authors have addressed all of my comments and concerns. The quality of the manuscript is improved and I recommend that it is published.

Reviewer #1 (Remarks on code availability):

I downloaded the code and run it in Matlab. The code is used to generate the results shown in the paper. However, these Matlab files use a data file (8.mat) where the raw data is stored. The raw data file is not included online and should be included.

Reviewer #2 (Remarks to the Author):

I thank the authors for this review. They have addressed my concerns.

Reviewer #2 (Remarks on code availability):

I briefly reviewed this code. This is Matlab code to regenerate the figures of the paper. Since I do not have a Matlab license, I could not dive further. It looks reasonable from first glance.